# Geometry-Aware Image Flow Matching

**Junho Lee** [* 1]  **Kwanseok Kim** [* 1 2]  **Joonseok Lee** [1]

## Abstract

Recent advances in generative models highlight the power of geometry-aware modeling in manifold-constrained settings. Yet, for natural images, the field remains confined to Euclidean assumptions, failing to exploit the potential of intrinsic geometric structures within the data. In this work, we investigate the geometry of natural images and observe that semantic information is predominantly encoded in directional components, while norm components can be approximated by the global average. This property holds across both RGB and latent spaces, suggesting that natural images can be effectively modeled on a hypersphere. Building on this finding, we introduce Spherical Optimal Transport Flow Matching (SOT-CFM), which utilizes angular distance, and Spherical Flow Matching (SFM), which constrains dynamics directly on the manifold. Our experiments demonstrate that these geometry-aware methods achieve superior performance against Euclidean baselines. Ultimately, this work provides a novel perspective that bridges the gap between Riemannian manifold-based modeling and natural image generation.

## 1. Introduction

Image generation has seen rapid progress through successive paradigms, from Continuous Normalizing Flows (CNF) (Chen et al., 2018; Grathwohl et al., 2018) to Diffusion models (DM) (Song & Ermon, 2019; Sohl-Dickstein et al., 2015; Ho et al., 2020; Song et al., 2021; Dhariwal & Nichol, 2021; Karras et al., 2022), and more recently to Flow Matching (FM) (Lipman et al., 2023; Liu et al., 2022; Albergo et al., 2023) approaches. Each breakthrough has delivered increasingly impressive results in terms of sample quality, training stability, and generation efficiency.

However, despite these advances, all these methods fundamentally rely on Euclidean geometry assumptions, treating images as vectors in high-dimensional Euclidean space. While this approach has proven successful, it may not fully capture the intrinsic geometric structure of natural images. If we could better understand and leverage the geometry of image data, we might achieve more principled and effective generative modeling.

In domains where the underlying data manifold is known, geometry-aware generative modeling has delivered tangible gains. Early work on Riemannian CNF (Mathieu & Nickel, 2020) parameterizes flexible densities directly on smooth manifolds by integrating ODEs on the manifold. Subsequent Riemannian score-based (Bortoli et al., 2022) and Riemannian Diffusion models (Huang et al., 2022) generalized score estimation and diffusion samplers to arbitrary Riemannian manifolds by formulating score operators and diffusion processes with Riemannian gradients/divergences. More recently, Riemannian Flow Matching (RFM) (Chen & Lipman, 2024) mitigates simulator bias by aligning geodesic velocities with closed-form target vector fields on Riemannian manifolds. In application domains where geometry is dictated by symmetries (*e.g.*, periodic crystals), FlowMM (Miller et al., 2024) extends RFM with group-equivariant structure, reporting state-of-the-art structure generation with substantially fewer integration steps. Collectively, these methods exploit geodesics, parallel transport, and manifold-aware metrics to obtain higher-quality samples, faster convergence, and more principled training relative to Euclidean baselines when the geometric prior is correct. However, a fundamental challenge remains: unlike structured domains with well-understood geometric priors, the intrinsic manifold structure of natural images is largely unknown. Although prior work has characterized their statistical properties (Ruderman, 1994) and local behavior (Carlsson et al., 2008), these insights have not yet been translated into exploitable geometric structures for generative modeling. Explicit geometric constraints or symmetries that could define a Riemannian manifold for images remain undiscovered.

To bridge this gap, we investigate the intrinsic geometry of natural images through directional decomposition analysis. Our key insight is that semantic information is predominantly encoded in the directional component (unit vector),

---
[*]Equal contribution  [1]Seoul National University, Seoul, Korea [2]TwelveLabs, Seoul, Korea. Correspondence to: Joonseok Lee <joonseok@snu.ac.kr>.

*Proceedings of the $43^{rd}$ International Conference on Machine Learning*, Seoul, South Korea. PMLR 306, 2026. Copyright 2026 by the author(s).

whereas magnitude (norm) contributes minimally to perceptual quality. Consequently, the norm of individual data points can be effectively approximated by the global average of the dataset. Crucially, while this property might seem intuitive in RGB space, we demonstrate that it holds true even for high-dimensional latent spaces optimized for reconstruction.

As illustrated in Figure 1, hyperspherical projection preserves semantic and visual integrity so effectively that the projected versions remain nearly indistinguishable from the originals, despite substantial changes in their $L_2$ norms across both RGB and latent spaces. This observation suggests that natural images can be effectively regarded as data points lying on a hypersphere with a radius determined by the dataset's average magnitude, both in RGB and latent spaces.

This finding enables us to establish geometry-aware image flow matching by either projecting data onto hyperspheres to leverage spherical geometry or by utilizing directional metrics instead of Euclidean metrics between vectors. Beyond its geometric benefits, this spherical projection also provides a practical training advantage by simplifying the learning task—since all projected data points reside on the same sphere with a predetermined radius, models can focus exclusively on learning directional dynamics rather than jointly optimizing both direction and magnitude components. Leveraging these insights, we introduce two approaches that adapt existing flow matching methods to spherical geometry: Spherical Optimal Transport Conditional Flow Matching (SOT-CFM), which replaces Euclidean distances with angular metrics in OT-CFM (Tong et al., 2024; Pooladian et al., 2023) for optimal transport coupling and Spherical Flow Matching (SFM), which operates entirely on the hyperspherical manifold by projecting both source and target distributions onto the sphere and using geodesic paths as the optimal transport trajectories between data points.

We validate our geometric approach through comprehensive experiments on CIFAR-10 and ImageNet-256. Specifically, we benchmark standard Euclidean baselines (I-CFM, OT-CFM) against our spherical adaptations of these frameworks. We observe that applying our spherical data projection relieves the burden of magnitude modeling, effectively lowering learning difficulty and directly translating to improved generation quality. Furthermore, SOT-CFM gains additional advantages through angular distance metrics. Most notably, SFM achieves the best performance among all evaluated variants. This work offers a novel perspective as the first successful application of Riemannian manifold-based generative methods to natural images, demonstrating the superior efficacy of intrinsic geometric modeling over standard Euclidean approaches.

The key contributions of this work are threefold:

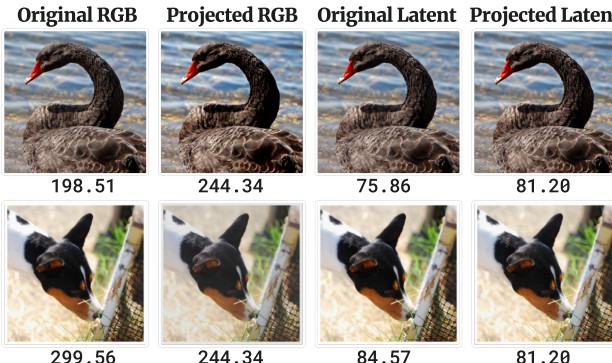

*Figure 1.* **Effect of hyperspherical projection on RGB and latent representations.** Original (columns 1, 3) and projected (columns 2, 4) versions in RGB and latent spaces. L2 norms below each image. Despite appreciable norm changes through projection, the images remain almost indistinguishable to human perception. Images are shown in the original pixel value range without per-image normalization.

- We discover and empirically validate that natural images exhibit an intrinsic *hyperspherical manifold structure*, where semantic information is dominantly encoded in directions.

- Leveraging this finding, we propose two *geometry-aware flow matching frameworks*—SOT-CFM and SFM—that unlock the potential of spherical manifold modeling for natural images

- This work establishes the *first practical bridge between manifold-based generative modeling and natural images*, enabling simple yet principled spherical geometric constraints to be effectively utilized in real-world visual domains.

## 2. Preliminary

### 2.1. Flow Matching (FM)

Flow Matching (FM) is a generative modeling framework defined on $\mathbb{R}^d$, where $d$ denotes the data dimensionality. It transports a source distribution $p_0$ (typically standard Gaussian) to a target data distribution $p_1$ via a time-dependent velocity field. Let $x_0 \sim p_0$ and $x_1 \sim p_1$ denote samples from the source and target distributions, respectively. Let $\{p_t\}_{t \in [0,1]}$ be a family of intermediate distributions interpolating between $p_0$ and $p_1$, and let $u_t : \mathbb{R}^d \times [0,1] \to \mathbb{R}^d$ denote a (marginal) velocity field that generates this path. Formally, the random process $(x_t)_{t \in [0,1]}$ obeys the ODE

$$\frac{dx_t}{dt} = u_t(x_t, t), \qquad x_t \sim p_t. \tag{1}$$

Flow Matching learns a neural approximation $v_\phi : \mathbb{R}^d \times [0,1] \to \mathbb{R}^d$ to the true marginal velocity field $u_t$ by mini-

mizing the squared error:

$$\min_{\phi} \; \mathbb{E}_{t\sim\mathcal{U}[0,1],\, x_t\sim p_t} \left[ \left\| v_\phi(x_t,t) - u_t(x_t,t) \right\|^2 \right]. \quad (2)$$

At sampling time, the learned vector field $v_\phi$ is used to approximately transport $p_0$ to $p_1$ by solving the ODE

$$\frac{dx_t}{dt} = v_\phi(x_t,t), \qquad x_0 \sim p_0. \quad (3)$$

However, the marginal velocity $u_t(x_t) = \mathbb{E}[u_t(x_t|x_0,x_1)|x_t]$ is intractable to compute directly. Conditional Flow Matching (CFM) (Lipman et al., 2023; Liu et al., 2022; Albergo et al., 2023) circumvents this by constructing a conditional probability path $p_t(x|x_0,x_1)$ and matching the conditional velocity field:

$$\mathcal{L}_{\text{CFM}}(\phi) = \mathbb{E}_{t,(x_0,x_1)\sim\pi,x_t\sim p_t(\cdot|x_0,x_1)} \left[ \|v_\phi(x_t,t) - u_t(x_t|x_0,x_1)\|^2 \right], \quad (4)$$

where $\pi$ is a coupling between $p_0$ and $p_1$ with marginals $(x_0,x_1)\sim\pi$ satisfying $x_0 \sim p_0$ and $x_1 \sim p_1$.

The standard choice is the linear interpolation path:

$$x_t = (1-\alpha(t))x_0 + \alpha(t)x_1 + \sigma(t)\varepsilon, \quad \varepsilon \sim \mathcal{N}(0,I), \quad (5)$$

where $\alpha : [0,1] \to [0,1]$ is an interpolation schedule with $\alpha(0) = 0$ and $\alpha(1) = 1$, and $\sigma(t)$ controls the noise level. The corresponding conditional velocity field has a closed-form expression:

$$u_t(x_t|x_0,x_1) = \dot{\alpha}(t)(x_1 - x_0) + \dot{\sigma}(t)\varepsilon. \quad (6)$$

When $\sigma(t) \equiv 0$, this reduces to the deterministic case with $u_t = \dot{\alpha}(t)(x_1 - x_0)$. Common choices include the linear schedule $\alpha(t) = t$ for Optimal Transport (OT) paths and more general schedules for improved sample quality.

## 2.2. Optimal Transport CFM (OT-CFM)

Standard Conditonal Flow Matching uses independent coupling between source and target distributions, known as Independent Conditional Flow Matching (I-CFM), which can result in inefficient transport paths. Optimal Transport Conditional Flow Matching (OT-CFM) (Tong et al., 2024; Pooladian et al., 2023) addresses this by finding optimal pairings between source and target points using optimal transport theory.

Instead of independent sampling from $p_0$ and $p_1$, OT-CFM solves the optimal transport problem:

$$\min_{\pi\in\Pi(p_0,p_1)} \mathbb{E}_{(x_0,x_1)\sim\pi}[c(x_0,x_1)], \quad (7)$$

where $c(x_0,x_1)$ is a cost function (typically $\|x_0 - x_1\|^2$) and $\Pi(p_0,p_1)$ denotes the set of all joint distributions with marginals $p_0$ and $p_1$. In practice, the exact optimal coupling

$\pi$ cannot be computed for entire dataset, so mini-batch optimal transport approximation is employed, where the optimal coupling is computed only over finite mini-batches. This approach creates simpler flows with straighter trajectories that are more stable to train and enable faster inference.

## 2.3. Riemannian Flow Matching (RFM)

While standard Flow Matching operates in Euclidean space, many applications benefit from incorporating geometric structure. Riemannian Flow Matching (RFM) (Chen & Lipman, 2024) extends flow matching to Riemannian manifolds $\mathcal{M}$ equipped with a metric tensor $g$. On a Riemannian manifold, the flow evolves according to:

$$\frac{dx}{dt} = u_t(x_t), \quad x_t \in \mathcal{M} \quad (8)$$

where $u_t(x) \in T_x\mathcal{M}$ is a time-dependent vector field in the tangent space at $x$.

The key innovation of RFM is constructing conditional vector fields using geodesics, the shortest paths on the manifold. For a conditional flow from $x_0$ to $x_1$, the conditional vector field is defined as

$$u_t(x_t|x_0,x_1) = \frac{d}{dt}\gamma_t(x_0,x_1), \quad (9)$$

where $\gamma_t(x_0,x_1)$ is the geodesic connecting $x_0$ and $x_1$, parameterized by $t \in [0,1]$.

The RFM training objective is given by

$$\mathcal{L}_{\text{RFM}}(\phi) = \mathbb{E}_{t,x_0,x_1,x_t} \left[ \|v_\phi(x_t,t) - u_t(x_t|x_0,x_1)\|_g^2 \right], \quad (10)$$

where $\|\cdot\|_g$ denotes the Riemannian norm induced by the metric $g$.

# 3. Flow matching on Spherical Geometry

In Section 2, we formalize the framework of Flow Matching and Riemannian Flow Matching (RFM). While previous works demonstrate that RFM yields tangible gains by leveraging known geometric priors, extending this success to image generation presents a fundamental challenge. As discussed in Section 1, the lack of a known intrinsic manifold for natural images makes it difficult to define geodesics, leaving existing geometry-aware methods largely inapplicable to image generation.

In this section, we address this limitation by discovering geometric structure within the data itself. Our approach centers on a key insight: through analysis of directional and norm decomposition, we demonstrate that natural images can be effectively approximated by spherical geometry. This foundational finding unlocks the capability to apply geometry-aware frameworks to natural image generation.

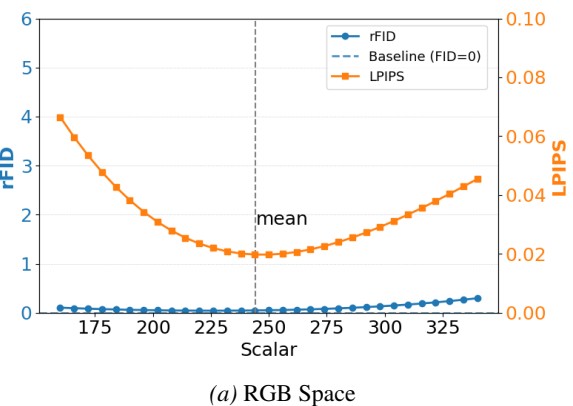

*(a)* RGB Space

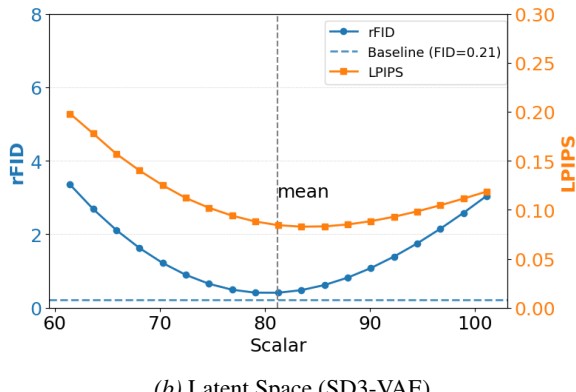

*(b)* Latent Space (SD3-VAE)

*Figure 2.* **Robustness analysis of spherical projection on ImageNet-256.** We evaluate reconstruction quality using rFID and LPIPS metrics after projecting data onto hyperspheres of varying radii (norms). The analysis is conducted in **(a)** pixel space (RGB) and **(b)** the latent space of SD3-VAE (Esser et al., 2024). The vertical dashed line indicates the global average norm of the dataset. The results show stable performance across a wide range of radii around the mean, suggesting that semantic content is effectively preserved in the directional components.

### 3.1. Vector Decomposition and Directional Analysis

To understand the geometric structure underlying image data, we treat each image as a flattened vector in $\mathbb{R}^d$ and begin with a basic observation: any such vector can be decomposed into its directional and norm components. Formally, given an image vector $x \in \mathbb{R}^d$, we can express it as:

$$x = \|x\|_2 \cdot \frac{x}{\|x\|_2} = s \cdot \hat{x}, \quad (11)$$

where $s = \|x\|_2$ is the magnitude (norm) and $\hat{x} = x/\|x\|_2$ is the unit direction vector lying on the $(d-1)$-dimensional unit hypersphere $\mathbb{S}^{d-1}$.

We note that $\hat{x}$ naturally resides on the unit hypersphere $\mathbb{S}^{d-1}$. This property is critical because if the magnitudes $s$ were approximately homogeneous, the image manifold could be directly approximated as a sphere.

To explore this hypothesis, we project image datasets onto hyperspheres of varying radii in both RGB and various latent spaces, preserving only the directional components while modifying the norm. We then measure reconstruction quality using rFID and LPIPS metrics compared to the original dataset. As shown in Figure 2a, rFID remains near zero across a wide range of radii in RGB space and LPIPS stays consistently low. Remarkably, we observe similar robustness patterns in the latent space of SD3-VAE (Esser et al., 2024) at Figure 2b.

Figure 1 provides visual confirmation of this robustness. Even with significant norm changes induced by projection, the visual differences remain nearly imperceptible. This quality persists across both RGB and latent spaces. Furthermore, our findings demonstrate that this property is not specific to a single setting; it extends to multiple autoencoder latent spaces within the LDM framework (see Table 1) and generalizes across datasets such as CIFAR-10, ImageNet, COCO-2014, and CelebA-HQ (Section D.1).

These findings show that most of the meaningful information lies in the directional component, while the norm component can be well-approximated by a global average. Based on this observation, we can project all data onto a single hypersphere, which offers significant advantages for generative modeling. First, by eliminating the need to match norms, the model can dedicate its entire capacity to learning the semantically important directional variations, reducing training complexity. Second, this spherical projection naturally enables *geometry-aware image generative modeling* by providing an explicit geometric structure to exploit.

### 3.2. Spherical OT-CFM with Angular Metrics

The observation that image semantics are primarily encoded in directional components provides a natural motivation to revisit OT-CFM (Tong et al., 2024; Pooladian et al., 2023) through the lens of spherical geometry. Standard OT-CFM constructs pairings by minimizing a Euclidean transport cost, implicitly assuming that both the direction and magnitude of image vectors carry comparable semantic meaning. However, our analysis in Section 3.1 shows that this assumption does not hold for natural images: direction captures the dominant semantic content, while magnitude mainly reflects low-level intensity variations.

This mismatch becomes evident when comparing pairs $(x_0, x_1)$ that share the same angular separation $\theta$—and are therefore semantically similar—but differ in magnitude. The Euclidean cost decomposes as:

$$\|x_0 - x_1\|^2 = \|x_0\|^2 + \|x_1\|^2 - 2\|x_0\|\|x_1\|\cos\theta. \quad (12)$$

This decomposition reveals that even with identical $\theta$,

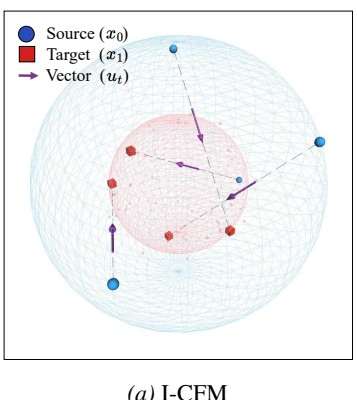

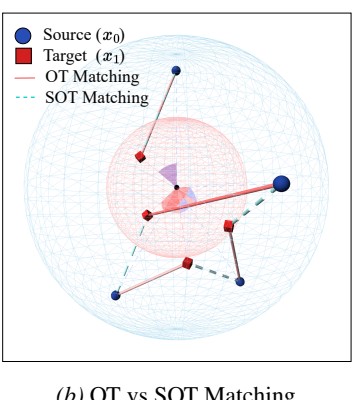

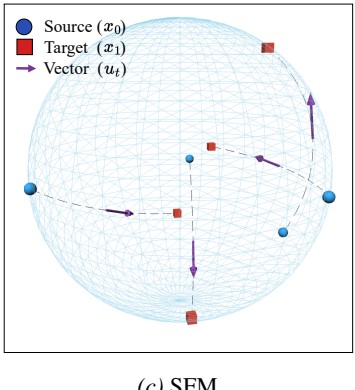

*(a)* I-CFM    *(b)* OT vs SOT Matching    *(c)* SFM

*Figure 3.* **Comparison of different flow matching strategies.** (a) I-CFM: Samples from a prior distribution ($x_0$, blue circles ●) are randomly paired with data samples ($x_1$, red squares ■) via straight-line paths in Euclidean space. (b) OT vs. SOT Matching: Standard OT (solid red —) minimizes Euclidean distance, potentially creating pairings with large angular differences. SOT (dashed cyan ---) matches points by angular proximity, better preserving semantic structure. (c) SFM: Flows are constrained to a spherical manifold with geodesic paths (great-circle arcs) between samples and tangent vector fields ($u_t$).

the cost is heavily influenced by magnitude differences ($\|x_0\|, \|x_1\|$). Since magnitude carries little semantic information, Euclidean OT may assign high costs to semantically similar pairs, leading to suboptimal and inconsistent matchings.

Angular distance, in contrast, directly compares the directional components of image vectors—the part of the representation where semantic information actually resides—while discarding magnitude variations that carry far weaker semantic signal. Motivated by this, we introduce Spherical OT-CFM (SOT-CFM), which replaces the Euclidean transport cost with an angular metric operating on the directional components of the data:

$$c_{\text{ang}}(x_0, x_1) = \arccos\left(\frac{\langle x_0, x_1 \rangle}{\|x_0\|_2 \|x_1\|_2}\right). \quad (13)$$

This angular cost is invariant to magnitude differences, ensuring that the optimal transport plan prioritizes semantic similarity and yields geometry-consistent couplings aligned with the intrinsic structure of natural images.

With the angular cost, the optimal transport problem in SOT-CFM becomes:

$$\min_{\pi \in \Pi(p_0, p_1)} \mathbb{E}_{(x_0, x_1) \sim \pi} \left[ \arccos\left(\frac{\langle x_0, x_1 \rangle}{\|x_0\|_2 \|x_1\|_2}\right) \right], \quad (14)$$

where $\Pi(p_0, p_1)$ denotes the set of all couplings between $p_0$ and $p_1$. Figure 3 (b) conceptually illustrates the difference between OT pairing and SOT pairing by showing how SOT-CFM matches samples along the directional component on the sphere. This reformulation naturally respects the spherical structure of the data manifold and offers several key advantages for image generation. By focusing optimization on the semantically meaningful directional manifold, it

provides better alignment with human perception of visual similarity.

### 3.3. Spherical Flow Matching

While SOT-CFM addresses transport cost issues by replacing Euclidean distance with angular distance, the spherical nature of image data can be leveraged more directly. Rather than only modifying the coupling strategy, we propose Spherical Flow Matching (SFM), which constrains both source and target distributions to the hypersphere manifold $\mathbb{S}^{d-1}$ and defines flow paths as geodesics on the manifold, allowing the entire flow dynamics to operate within the spherical geometry.

This approach is well-motivated by two complementary observations. First, high-dimensional Gaussian noise effectively lie on or near the surface of a hyperspherical shell. Specifially, the $L_2$ norm of Gaussian noise $x_0 \sim \mathcal{N}(0, I_d)$ follows a $\chi$-distribution with $d$ degrees of freedom, whose mean and variance asymptotically converge to $\sqrt{d - \frac{1}{2}}$ and $\frac{1}{2}$, respectively, as $d$ becomes large. This concentration phenomenon causes the radii of high-dimensional Gaussian samples to cluster tightly around their expected value as dimensionality increases. Second, most meaningful information in images resides in the directional component, while the norm can be well-approximated by a global average as discussed in Section 3.1. Leveraging these two properties, we project both the source Gaussian distribution and the target image data onto the same hypersphere, enabling the flow dynamics to operate purely within this directional geometric space, as illustrated in Figure 3(c). This allows our model to focus computational resources on learning the directionally meaningful variations.

On the hypersphere, the shortest path connecting any two points is the geodesic, which has a closed-form expression as spherical linear interpolation (slerp):

$$\tilde{x}_t = \gamma_t(\tilde{x}_0, \tilde{x}_1) = \frac{\sin((1-t)\theta)}{\sin\theta}\tilde{x}_0 + \frac{\sin(t\theta)}{\sin\theta}\tilde{x}_1, \quad t \in [0,1], \quad (15)$$

where $\tilde{x}_0 = r \cdot x_0/\|x_0\|_2$ and $\tilde{x}_1 = r \cdot x_1/\|x_1\|_2$ are the projected vectors on the hypersphere of radius $r$, and $\theta = \arccos(\langle\tilde{x}_0, \tilde{x}_1\rangle/r^2)$ is the angle between them. See Section A for a detailed derivation.

Following the geodesic path, we can derive the conditional vector field $u_t$ at any point $\tilde{x}_t$ along the trajectory. By construction, this vector field $u_t$ is always in the tangent space $T_{x_t}\mathbb{S}^{d-1}$ for all $t \in [0,1]$. Our goal is to train a model $v_\phi(t, x_t)$ to predict this tangent vector. Unlike Euclidean flow matching, SFM measures the discrepancy using the Riemannian inner product induced by the hypersphere geometry. Specifically, for a base point $x_t$ and tangent vectors $u, v \in T_{x_t}\mathbb{S}^{d-1}$, the inner product is:

$$\langle u, v \rangle_{g(\tilde{x}_t)} = u^\top v, \quad (16)$$

since the hypersphere inherits the Euclidean metric restricted to the tangent space. The SFM loss is thus formulated as:

$$\mathcal{L}_{\text{SFM}}(\phi) = \mathbb{E}_{t, \tilde{x}_0, \tilde{x}_1, \tilde{x}_t}\left[\|v_\phi(t, \tilde{x}_t) - u_t(\tilde{x}_t|\tilde{x}_0, \tilde{x}_1)\|^2\right]. \quad (17)$$

By optimizing this geometrically-grounded loss, SFM effectively constrains the entire generative process to the hypersphere, where crucial semantic information resides. This approach establishes the first practical application of manifold-based generative modeling to natural images. It demonstrates the viability of geometry-aware frameworks for real-world image generation and provides a foundation for future models exploiting the intrinsic geometric structure of natural data.

## 4. Experiment

### 4.1. Experimental Setup

**Datasets.** We conduct experiments on two standard image generation benchmarks: CIFAR-10 (Krizhevsky & Hinton, 2009) and ImageNet-256 (Russakovsky et al., 2015). CIFAR-10 consists of 50,000 training images across 10 classes at $32 \times 32$ resolution, while ImageNet-256 contains approximately 1.28 million training images from 1,000 classes at $256 \times 256$ resolution. For ImageNet-256, we perform class-conditional generation to evaluate our methods' ability to incorporate semantic conditioning.

**Evaluation Metrics.** We evaluate all methods using standard generative modeling metrics computed on 50,000 generated samples. We report Generative Fréchet Inception

Distance (gFID) (Heusel et al., 2017) to measure the distributional distance between generated and real images, sFID (Nash et al., 2021), a variation of FID using spatial features, better captures spatial relationships and high-level structure in image distributions, Inception Score (IS) (Salimans et al., 2016) to evaluate sample quality and diversity, and Precision and Recall (Nichol & Dhariwal, 2021) to assess fidelity and coverage of the generated distribution.

**Baselines.** We evaluate our spherical adaptations against the standard Euclidean baselines within two established frameworks: I-CFM and OT-CFM. Additionally, we evaluate Spherical Flow Matching (SFM) to demonstrate the efficacy of our fully Riemannian framework. This comparison quantifies two distinct benefits: training efficiency gains from spherical data projection and performance improvements from intrinsic geometric modeling.

**Inference Configuration.** For CIFAR-10, we perform unconditional generation with a standard first-order Euler solver with 100 function evaluations (NFE). For ImageNet-256, we perform class-conditional generation with classifier-free guidance (CFG), using scales individually optimized for peak performance (2.1 for I-CFM, 2.6 for OT-CFM/SOT-CFM, and 2.3 for SFM). Sampling is performed using an Euler ODE solver with 250 steps. All quantitative metrics reported in Table 2 are computed with 50,000 generated samples.

| Space | rFID | | LPIPS | |
|---|---|---|---|---|
| | Baseline | Mean Projected | Baseline | Mean Projected |
| RGB | 0 | 0.05 (+0.05) | 0 | 0.02 (+0.02) |
| SD2-VAE | 0.71 | 1.14 (+0.43) | 0.13 | 0.15 (+0.02) |
| SD3-VAE | 0.21 | 0.40 (+0.19) | 0.06 | 0.08 (+0.02) |
| VMAE | 0.89 | 0.88 (−0.01) | 0.06 | 0.06 (±0.00) |
| DC-AE | 1.02 | 1.57 (+0.55) | 0.17 | 0.18 (+0.01) |

*Table 1.* **Impact of hyperspherical projection on reconstruction quality.** We compare reconstruction metrics (rFID and LPIPS) between original data (Baseline) and data projected onto hyperspheres using mean norm (Mean Projected) across different representation spaces on ImageNet-256. Values in parentheses indicate changes from baseline.

### 4.2. Analysis of Representation Spaces

Before presenting our main results, we validate our core hypothesis about the spherical nature of image data across different representation spaces. Table 1 shows the impact of projecting datasets onto hyperspheres with global average norm $\bar{s}$ while preserving directional components. We evaluate reconstruction quality using rFID (FID between original and projected data) and LPIPS metrics. The results confirm our hypothesis that across RGB space and multiple autoencoder latent spaces (SD2-VAE (Rombach et al., 2022), SD3-VAE (Esser et al., 2024), VMAE (Lee et al., 2025), DC-AE (Chen et al., 2024)), projecting data onto a hypersphere with average radius preserves most semantic

| Method | Source | Target | CIFAR10 | Class-Conditional ImageNet256 | | | | |
|--------|--------|--------|---------|---------|---------|---------|---------|---------|
| | | | gFID↓ | gFID↓ | sFID↓ | IS↑ | Precision↑ | Recall↑ |
| I-CFM | $\mathcal{N}$ | $\mathcal{D}$ | 4.29 | 5.29 | 8.24 | 236.37 | 0.8250 | 0.4604 |
| I-CFM (ours) | $\mathcal{N}$ | $\tilde{\mathcal{D}}$ | 4.10 | 5.02 | 7.98 | 236.13 | 0.8073 | 0.4827 |
| OT-CFM | $\mathcal{N}$ | $\mathcal{D}$ | 4.30 | 5.22 | 7.85 | 247.71 | **0.8258** | 0.4528 |
| SOT-CFM (ours) | $\mathcal{N}$ | $\mathcal{D}$ | 4.11 | 5.15 | 7.80 | 235.99 | 0.8181 | 0.4654 |
| SOT-CFM (ours) | $\mathcal{N}$ | $\tilde{\mathcal{D}}$ | 4.11 | 5.10 | 7.69 | 234.76 | 0.8128 | 0.4611 |
| SFM | $\mathcal{N}$ | $\mathcal{D}$ | *inapplicable* | | | | | |
| SFM (ours) | $\tilde{\mathcal{N}}$ | $\tilde{\mathcal{D}}$ | **3.79** | **4.62** | **7.51** | **337.85** | 0.8083 | **0.4890** |

*Table 2.* **Generative performance comparison on CIFAR-10 and ImageNet-256.** $\mathcal{N}$ and $\tilde{\mathcal{N}}$ denote the Gaussian and spherically projected Gaussian distributions (which is nearly identical to uniform distribution on the sphere), respectively. $\mathcal{D}$ and $\tilde{\mathcal{D}}$ denote the original and spherically projected datasets, respectively. SFM is inherently designed for spherical manifolds and cannot be directly applied to Euclidean data.

information. Notably, the SD3-VAE latent space shows particularly robust behavior with only a 0.20 increase in rFID, while VMAE demonstrates near-perfect preservation with a negligible $-0.01$ change. Additionally, LPIPS maintains near-baseline performance across all representation spaces. We further substantiate the generalizability of these findings through extensive experiments on additional datasets, as detailed in Section D.1. This collective evidence validates our assumption that directional information dominates semantic content across diverse representation spaces.

### 4.3. Quantitative Comparison

We present our main experimental results in Table 2, comparing our proposed spherical methods against Euclidean baselines on the CIFAR-10 and ImageNet-256 datasets.

**Utilizing Spherical Geometry.** Projecting data onto a hypersphere leads to consistent and meaningful improvements across both CIFAR-10 and ImageNet-256. These gains occur without modifying the model architecture, sampling strategy, or training objective—indicating that the improvement arises purely from the spherical projection.

By removing norm variability, spherical projection simplifies the learning problem focusing on the semantic content encoded in directions: all samples lie on a common hypersphere with consistent scale. This allows the model to focus on directional semantic structure without being distracted by magnitude fluctuations. Thus, the projection results empirically validate our hypothesis that projecting representations onto a fixed-radius hypersphere preserves their semantic content and focuses the model on the semantically meaningful part of the representation, thereby improving both stability and generative quality.

Furthermore, our SOT-CFM, which replaces Euclidean transport costs with angular distances to better capture directional similarity, yields consistent improvements over OT-CFM. On CIFAR-10, it achieves a gFID of 4.11 compared to

OT-CFM's 4.30, and on ImageNet-256, its gFID improves from 5.22 to 5.15. While these gains are relatively small, they appear reliably across datasets. Combining spherical projection with SOT-CFM maintains similar performance on CIFAR-10 and further improves ImageNet-256 results, suggesting that projection and angular transport provide complementary benefits. These observations indicate that focusing on directional components leads to more semantically aligned pairings and incremental improvements in generative quality.

**Spherical Flow Matching.** Our SFM method, which defines both source and target distributions as well as flow paths directly on the hyperspherical manifold, achieves superior performance compared to all evaluated baselines. On CIFAR-10, SFM attains the best gFID of 3.79, consistently outperforming all Euclidean baselines including I-CFM, OT-CFM, and their spherically-projected variants.

Notably, SFM demonstrates robust performance on the more challenging ImageNet-256 dataset, achieving the best results across all metrics except precision. These comprehensive improvements demonstrate that operating directly on the Riemannian manifold provides tangible benefits over Euclidean methods, even when the latter are enhanced with spherical projections.

To our knowledge, this marks the first successful application of a fully Riemannian flow matching framework to large-scale natural image generation. Our results indicate that geometric tools from differential geometry are not merely theoretical constructs but practical alternatives that can surpass standard Euclidean methods, paving the way for further research into geometry-aware generative modeling.

### 4.4. Radius Ablation Study

Since SFM requires choosing the projection radius $r$, we analyze the effect of the hypersphere radius on generation quality. During training, both the data and Gaussian samples

| Radius ($r$) | 30 | 45.25 (Default) | 60 | 80 | 120 | 130 |
|---|---|---|---|---|---|---|
| gFID | 4.95 | 4.85 | 4.74 | 4.68 | **4.62** | 4.65 |

*Table 3.* **SFM across various radii on ImageNet-256.** We sweep the projection radius $r$ while keeping all other hyperparameters fixed. The default value (45.25) corresponds to the average $\ell_2$ norm of high-dimensional Gaussian samples at the latent dimensionality used in our experiments.

are projected onto a hypersphere of radius $r$; after generation, the produced samples are rescaled back to the original average norm $\bar{s}$ of the dataset. As shown in Table 3, performance consistently improves as $r$ increases, peaking at $r = 120$ (gFID: 4.62) before slightly degrading at $r = 130$, indicating a clear saturation regime.

This behavior admits a geometric interpretation. Recall that the conditional vector field in SFM is the geodesic velocity, whose magnitude is $\|\dot{\gamma}(t)\|_2 = r\theta$. Consequently, increasing $r$ directly scales the magnitude of tangent vectors in $T_{\tilde{x}_t}\mathbb{S}^{d-1}$ for a given angular displacement $\theta$, without altering the underlying directional structure of the flow. This amplification provides a stronger and less ambiguous regression target for the neural network, effectively improving the signal-to-noise ratio of the tangent space training objective. Beyond a certain radius, however, the benefit saturates as the increased tangent magnitude no longer meaningfully improves optimization conditioning, consistent with the slight degradation observed at $r = 130$.

Importantly, performance remains stable across a broad range of radii around the optimum, confirming that SFM is not sensitive to precise radius selection. This robustness is practically significant, as practitioners can simply set $r$ to a large value within a reasonable range without careful tuning. Furthermore, the consistent and monotonic improvement up to the saturation point strongly suggests that the gains stem from a genuine geometric effect rather than narrow hyperparameter tuning.

### 4.5. Qualitative Comparisons

Figure 4 presents side-by-side generated samples from I-CFM, I-CFM with hyperspherical projection (Ours), and SFM (Ours) on ImageNet-256. All methods share identical random seeds and class labels, so each row corresponds to the same noise input processed under each formulation.

I-CFM generates images that frequently exhibit structural incoherence and blurred fine-grained details, suggesting that jointly optimizing direction and magnitude introduces unnecessary learning complexity. Incorporating hyperspherical projection into the target distribution (I-CFM + Proj) alleviates this burden by constraining the model to focus on directional dynamics, yielding noticeably more coherent global structures. Our fully geometry-aware SFM takes this further by operating the entire generative process on

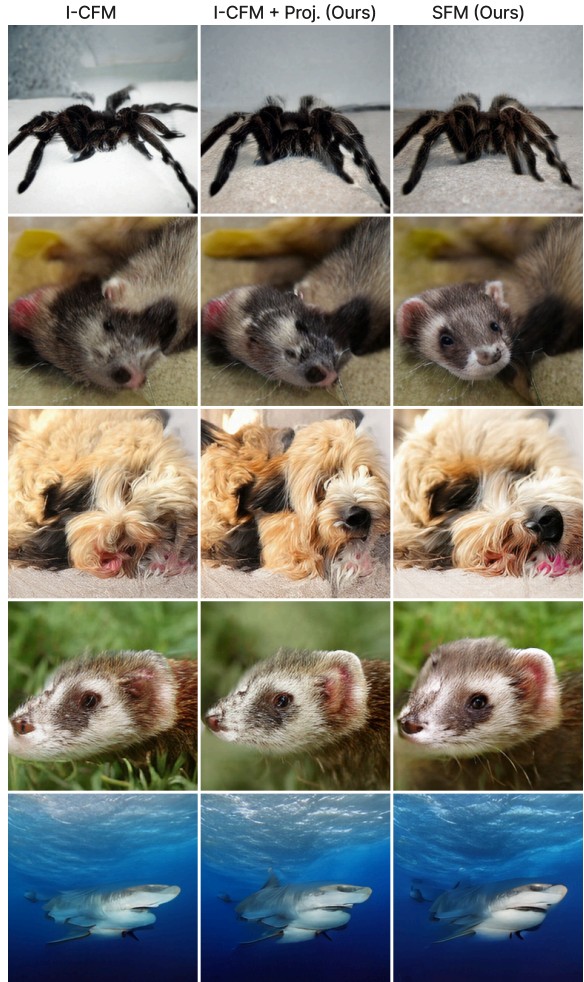

*Figure 4.* **Qualitative comparison of generated ImageNet-256 samples.** Each row shares the same noise seed and class label across methods. I-CFM (**left**) exhibits structural incoherence and blurred textures. I-CFM + Proj (**middle**) yields more coherent global structures. SFM (**right**) consistently produces the sharpest details and richest textures across diverse categories.

the hyperspherical manifold: the resulting samples display markedly sharper textures, well-defined fine-grained details, and stronger semantic fidelity across diverse ImageNet categories. These qualitative observations are consistent with the quantitative improvements reported in Table 2.

## 5. Related Work

### 5.1. Advances in Flow-Based Generative Models

While early Continuous Normalizing Flows (CNFs) (Chen et al., 2018; Grathwohl et al., 2018) pioneered invertible transformations for generative modeling, they suffered from expensive trace computations and training instability. Flow Matching (Lipman et al., 2023; Liu et al., 2022; Albergo et al., 2023) revolutionized this paradigm with simulation-free objectives that directly regress onto conditional vector

fields, dramatically improving scalability and enabling practical deployment.

Recent theoretical advances have expanded Flow Matching beyond linear interpolation to accommodate diverse probability paths. These include discrete state spaces for categorical data (Gat et al., 2024), Dirichlet paths for simplex-constrained distributions (Stark et al., 2024), alpha-blended trajectories for improved convergence (Cheng et al., 2025), and metric-aware paths that respect data geometry (Kapusniak et al., 2024).

The versatility of Flow Matching has driven breakthroughs across modalities. In visual domains, it powers state-of-the-art text-to-image synthesis (Esser et al., 2024), controllable generation (Dao et al., 2023), and AR applications (Ren et al., 2024). For temporal data, it enables high-fidelity audio (Guan et al., 2024; Liu et al., 2023) and music generation (Prajwal et al., 2024), as well as coherent video synthesis through hierarchical architectures (Jin et al., 2025; Polyak et al., 2024). Beyond media, Flow Matching excels in scientific applications including molecular docking (Dunn & Koes, 2024), equivariant conformer generation (Song et al., 2023), and discrete text generation (Hu et al., 2024).

## 5.2. Geometry-Aware Generative Modeling

Incorporating geometric priors has proven effective for data on known manifolds. Building on earlier work on normalizing flows for specific manifolds such as spheres and tori (Rezende et al., 2020), Riemannian Continuous Normalizing Flows (Mathieu & Nickel, 2020) provided a general framework integrating differential geometry into flow-based models via manifold-aware ODEs that respect local curvature. Similarly, Riemannian Score-Based (Bortoli et al., 2022) and Diffusion Models (Huang et al., 2022) extended diffusion processes to Riemannian manifolds using heat kernels and logarithmic maps, improving performance on structured data like molecular conformations. More recently, Riemannian Flow Matching (RFM) (Chen & Lipman, 2024) enabled simulation-free training on manifolds by matching geodesic velocities with closed-form expressions, and FlowMM (Miller et al., 2024) extended it with SE(3) group-equivariance for efficient crystal generation, achieving superior sample quality.

## 5.3. Optimal Transport in Generative Modeling

Optimal Transport (OT) provides theoretical foundations for generative models, notably used in Wasserstein GANs (Arjovsky et al., 2017) for stable adversarial training via Earth Mover's distance. In Flow Matching, OT offers principled pairings between source and target points (Tong et al., 2024; Pooladian et al., 2023), constructing minimal-cost transport maps that lead to shorter, straighter paths with improved sampling efficiency. However, existing OT methods often rely on Euclidean metrics which may not capture perceptual similarity in high-dimensional image data where directional or semantic distances could be more meaningful. Scalable algorithms like entropic-regularized Sinkhorn iterations (Cuturi, 2013) and progressive mini-batch solvers (Kassraie et al., 2024) have made OT computationally practical for large-scale generation tasks. Further refinements address challenges in conditional settings through class-aware transport penalties (Cheng & Schwing, 2025) or unbalanced partial optimal transport (Nguyen et al., 2022), while others learn adaptive coupling strategies directly from data distributions (Lin et al., 2025).

## 5.4. Autoencoders in the LDM Framework

Latent diffusion models rely on autoencoders to compress images into semantically meaningful, lower-dimensional latent spaces. The Autoencoder-KL from Stable Diffusion (Rombach et al., 2022) established this approach using perceptual and adversarial losses, which was later scaled up in SDXL (Podell et al., 2023) with higher-resolution training and Stable Diffusion v3 (Esser et al., 2024) incorporating rectified flows for improved perceptual fidelity and multimodal conditioning strategies. Beyond these, alternative encoders have been proposed to enhance latent representations and generation quality. VA-VAE (Yao et al., 2025) addresses the reconstruction-generation trade-off in latent diffusion by aligning the VAE latent space with CLIP vision embeddings from pretrained foundation models. MAEtok (Chen et al., 2025) and VMAE (Lee et al., 2025) combined latent diffusion with masked autoencoding pretraining, improving latent space structure and semantic consistency. In parallel, deep compression autoencoders (DC-AE) (Chen et al., 2024) were designed with hierarchical bottlenecks to achieve 128× spatial reduction while maintaining reconstruction fidelity.

## 6. Conclusion

In this work, we investigated the critical role of geometric structure in image generation. Our analysis of directional decomposition reveals that natural images are effectively modeled on a hypersphere, where semantic information is primarily encoded in directions while scalar norms can be approximated by dataset averages. Building on this insight, we introduced SOT-CFM and SFM, which leverage angular metrics and geodesic dynamics to ensure geometrically consistent generative paths. Empirical results on CIFAR-10 and ImageNet-256 demonstrate that these spherical approaches consistently outperform Euclidean baselines across diverse representation spaces, including RGB and modern autoencoder latents. Ultimately, our work establishes that exploiting the intrinsic spherical geometry of natural images provides tangible practical advantages, establishing a robust foundation for geometry-aware image generation.

## Acknowledgments

This work was also supported by Samsung Electronics, Youl-chon Foundation, National Research Foundation of Korea (NRF) grants (RS-2021-NR05515, RS-2024-00336576, RS-2023-0022663), and the Institute for Information & Communication Technology Planning & Evaluation (IITP) grants (RS-2022-II220264, RS-2024-00353131) funded by the Korean government.

## Impact Statement

This paper advances geometry-aware image generation through Riemannian manifold theory. Our contributions are foundational in nature and do not introduce new application domains. Broader societal implications, such as potential misuse of generative models, are shared across the field and not specific to our work. We do not anticipate any immediate negative societal consequences from the techniques proposed in this work.

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

# Appendix

## A. Geodesic Interpolation and Tangent Vector Derivation

We provide the mathematical derivations for the geodesic interpolation on a hypersphere and the corresponding tangent vector field used in Spherical Flow Matching (SFM).

Let $\tilde{x}_0, \tilde{x}_1 \in \mathbb{R}^d$ be two points on a hypersphere of radius $r > 0$, i.e., $\|\tilde{x}_0\|_2 = \|\tilde{x}_1\|_2 = r$. The geodesic (shortest path) between these two points on the sphere is characterized by the great-circle arc that connects them. The angle between the two points is defined as

$$\theta = \arccos\left(\frac{\langle \tilde{x}_0, \tilde{x}_1 \rangle}{r^2}\right), \quad \theta \in [0, \pi].$$

Assuming cases $\theta = 0$ ($\tilde{x}_0 = \tilde{x}_1$) or $\theta = \pi$ ($\tilde{x}_0 = -\tilde{x}_1$) are negligible in practice (a set of measure zero), we focus on *the range $\theta \in (0, \pi)$ where $\sin\theta \neq 0$ is guaranteed*.

We can express the geodesic interpolation $\gamma(t)$, also known as spherical linear interpolation (SLERP), as

$$\gamma(t) = \frac{\sin((1-t)\theta)}{\sin\theta}\,\tilde{x}_0 + \frac{\sin(t\theta)}{\sin\theta}\,\tilde{x}_1, \quad t \in [0,1]. \quad (18)$$

This curve satisfies $\gamma(0) = \tilde{x}_0$, $\gamma(1) = \tilde{x}_1$, and $\|\gamma(t)\|_2 = r$ for all $t$, thus remaining on the hypersphere.

Differentiating Eq. (18) with respect to $t$ yields the tangent (velocity) vector along the geodesic:

$$\dot{\gamma}(t) = \frac{\theta}{\sin\theta}\left[-\cos((1-t)\theta)\,\tilde{x}_0 + \cos(t\theta)\,\tilde{x}_1\right]. \quad (19)$$

We denote this velocity as the conditional vector field $u_t(\tilde{x}_t \mid \tilde{x}_0, \tilde{x}_1) = \dot{\gamma}(t)$, where $\tilde{x}_t = \gamma(t)$. By construction, $\langle \dot{\gamma}(t), \gamma(t) \rangle = 0$, ensuring that $u_t$ always lies in the tangent space $T_{\tilde{x}_t}\mathbb{S}_r^{d-1}$.

Moreover, the speed of motion along the geodesic is constant since

$$\|\dot{\gamma}(t)\|_2 = r\theta, \quad \forall t \in [0,1],$$

which confirms that the path corresponds to the great-circle geodesic on the sphere.

Using this closed-form tangent vector field, the SFM training objective can be expressed as

$$\mathcal{L}_{\mathrm{SFM}}(\phi) = \mathbb{E}_{t,\tilde{x}_0,\tilde{x}_1,\tilde{x}_t}\left[\|v_\phi(t, \tilde{x}_t) - u_t(\tilde{x}_t \mid \tilde{x}_0, \tilde{x}_1)\|_2^2\right], \quad (20)$$

where the inner product and norm are induced by the Euclidean metric restricted to the tangent space of the sphere, i.e., $\langle u, v \rangle_{g(\tilde{x}_t)} = u^\top v$ for all $u, v \in T_{\tilde{x}_t}\mathbb{S}_r^{d-1}$.

For numerical stability, when $\theta \to 0$, we use the first-order approximations $\sin(t\theta)/\sin\theta \approx t$, $\sin((1-t)\theta)/\sin\theta \approx$
$1 - t$, and $\theta/\sin\theta \approx 1$, under which Eqs. (18) and (19) naturally reduce to their Euclidean linear interpolation counterparts:

$$\gamma(t) \approx (1-t)\tilde{x}_0 + t\tilde{x}_1, \quad u_t \approx \tilde{x}_1 - \tilde{x}_0.$$

This confirms that Spherical Flow Matching generalizes Euclidean flow matching, smoothly transitioning to the standard case when the curvature approaches zero.

## B. Implementation Details

### B.1. Model Architecture

To account for the significant differences in image resolution and dataset scale, we utilize architectures optimized for each respective task.

**CIFAR-10.** We employ a standard U-Net architecture (Dhariwal & Nichol, 2021) following the configuration from OT-CFM (Tong et al., 2024). The model uses 128 base channels with multipliers [1, 2, 2, 2] across 4 resolution levels, employing self-attention at 16×16 resolution. We use 2 residual blocks per resolution level with GroupNorm and SiLU activations. Time embeddings are processed through sinusoidal positional encoding followed by a 2-layer MLP.

**ImageNet-256.** We adopt DC-AE (Chen et al., 2024) f64d128 configuration as the autoencoder, which compresses 256×256×3 images to 4×4×128 latent representations. For the generative model, we use DiT-XL/2 (Peebles & Xie, 2023) with 28 transformer blocks, hidden dimension of 1152, 16 attention heads, and patch size of 2. The model uses AdaLN-Zero for conditional control and learnable positional embeddings.

### B.2. Training Configuration

**CIFAR-10.** Models are trained for 200,000 iterations with batch size 512, using Adam optimizer (Kingma & Ba, 2014) with learning rate $2 \times 10^{-4}$ and default betas (0.9, 0.999). We apply exponential moving average (EMA) with decay 0.9999. Training is performed on a single NVIDIA A6000 GPU.

**ImageNet-256.** We train for 140,000 iterations with batch size 1,024, using AdamW optimizer (Loshchilov & Hutter, 2017) with learning rate $2 \times 10^{-4}$, weight decay 0, and $\beta = (0.9, 0.95)$. Following LightningDiT, we use gradient checkpointing for memory efficiency and mixed precision training with fp16. Training is conducted on 2 NVIDIA A100 40GB GPUs.

### B.3. Optimal Transport Computation

For both standard and spherical OT, we compute mini-batch optimal transport with batch size 128 for CIFAR-10 and 256 for ImageNet-256. We use the POT library (Flamary et al.,

| Method | Source | Target | Norm Pred. | gFID↓ |
|--------|--------|--------|------------|-------|
| I-CFM | $\mathcal{N}$ | $\tilde{\mathcal{D}}$ | - | 5.02 |
|  | $\mathcal{N}$ | $\tilde{\mathcal{D}}$ | ResNet50 | 4.98 |
|  | $\mathcal{N}$ | $\tilde{\mathcal{D}}$ | MobileNetV2 | 4.96 |
| SOT-CFM | $\mathcal{N}$ | $\tilde{\mathcal{D}}$ | - | 5.18 |
|  | $\mathcal{N}$ | $\tilde{\mathcal{D}}$ | ResNet50 | 5.12 |
|  | $\mathcal{N}$ | $\tilde{\mathcal{D}}$ | MobileNetV2 | 5.11 |
| SFM | $\tilde{\mathcal{N}}$ | $\tilde{\mathcal{D}}$ | - | 4.84 |
|  | $\tilde{\mathcal{N}}$ | $\tilde{\mathcal{D}}$ | ResNet50 | 4.81 |
|  | $\tilde{\mathcal{N}}$ | $\tilde{\mathcal{D}}$ | MobileNetV2 | 4.81 |

*Table I.* **Effect of norm refinement on generation quality.** Comparison of gFID scores with and without norm prediction networks on ImageNet-256. Marginal improvements confirm that simple average-norm projection captures essential information.

| None | ResNet50 | MobileNet v2 |
|------|----------|--------------|

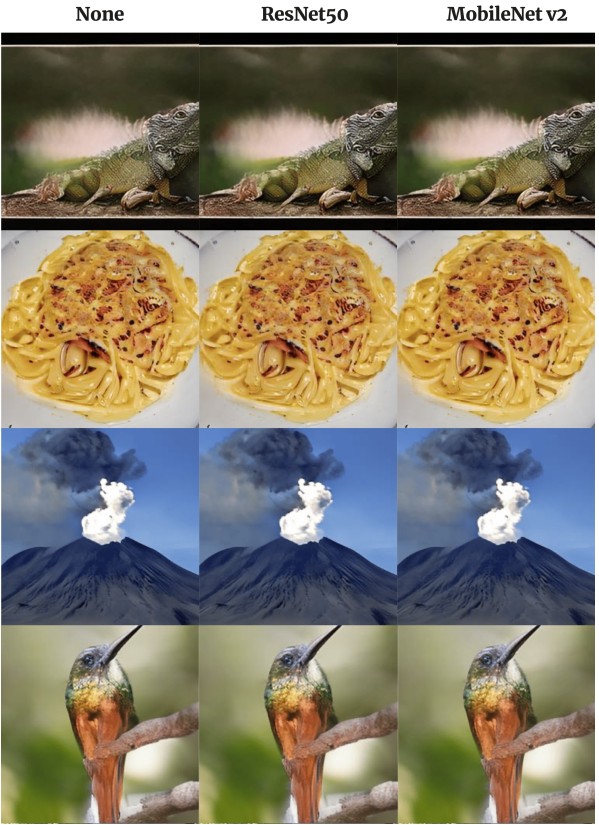

*Figure I.* **Visual comparison of generated samples with and without norm refinement.** Samples generated using I-CFM trained on projected ImageNet-256 comparing (left) no norm prediction, (middle) ResNet50-based refinement, and (right) MobileNetV2-based refinement. The indistinguishable visual quality suggests that the global average norm approximation is sufficiently robust, rendering additional adjustments like the Norm Refinement Network ($N_\phi$) unnecessary.

2021) with entropy regularization $\epsilon = 0.1$ and Sinkhorn iterations. For spherical OT, the cost matrix uses geodesic distance $c(x_0, x_1) = \arccos(\langle \hat{x}_0, \hat{x}_1 \rangle)$.

### B.4. Implementation Framework

Our implementation builds upon OT-CFM (Tong et al., 2024) for CIFAR-10 experiments and LightningDiT (Yao et al., 2025) for ImageNet-256 experiments. All code is implemented in PyTorch with reproducible seeds.

## C. Norm Prediction for Adjustment

Table 1 demonstrates that hyperspherical projection using the global average norm $\bar{s} = \frac{1}{N} \sum_{i=1}^{N} \|x_i\|_2$ effectively preserves essential information, though minor degradation remains observable. To investigate whether recovering this lost magnitude information improves generation quality, we experiment with a lightweight Norm Refinement Network ($N_\phi$) designed for fine-grained norm adjustments.

### C.1. Norm Refinement Network

We employ standard vision encoders—ResNet50 and MobileNetV2—to recover the magnitude information lost during projection. The network takes the unit direction vector $\hat{x}$ as input and predicts a norm correction term $\Delta s_\phi$, which approximates the true deviation $\Delta s = s - \bar{s}$. The final refined vector is then constructed as:

$$x_{\text{pred}} = (\bar{s} + \Delta s_\phi) \cdot \hat{x}. \qquad (21)$$

The network $N_\phi$ is optimized with a pixel-level reconstruction (MSE). For models operating directly in image space (e.g., RGB), the objective is:

$$\mathcal{L}_{\text{img}}(\phi) = \mathbb{E}_{x, x_{\text{pred}}} \left[ \|x - x_{\text{pred}}\|_2^2 \right]. \qquad (22)$$

For latent-space models, we apply the same refinement process to the unit latent vector $\hat{z}$, yielding $z_{\text{pred}} = (\bar{s} + \Delta s_\phi) \cdot \hat{z}$. The losses are computed after decoding through $\mathcal{D}$, with an additional latent consistency term:

$$\mathcal{L}_{\text{latent}}(\phi) = \mathbb{E}_{z, z_{\text{pred}}} \left[ \|x - \mathcal{D}(z_{\text{pred}})\|_2^2 + \lambda_1 \|z - z_{\text{pred}}\|_2^2 \right].$$

This refinement strategy is expected to enhance representations that are already well-approximated by the global average, further improving reconstruction fidelity while introducing negligible computational overhead at inference time.

### C.2. Quantitative and Qualitative Analysis

Table I presents the impact of norm refinement on generation quality. While the norm prediction networks consistently improve gFID scores across all methods and architectures, the gains are marginal. This minimal enhancement validates our core hypothesis: hyperspherical projection with average norm effectively preserves the essential information

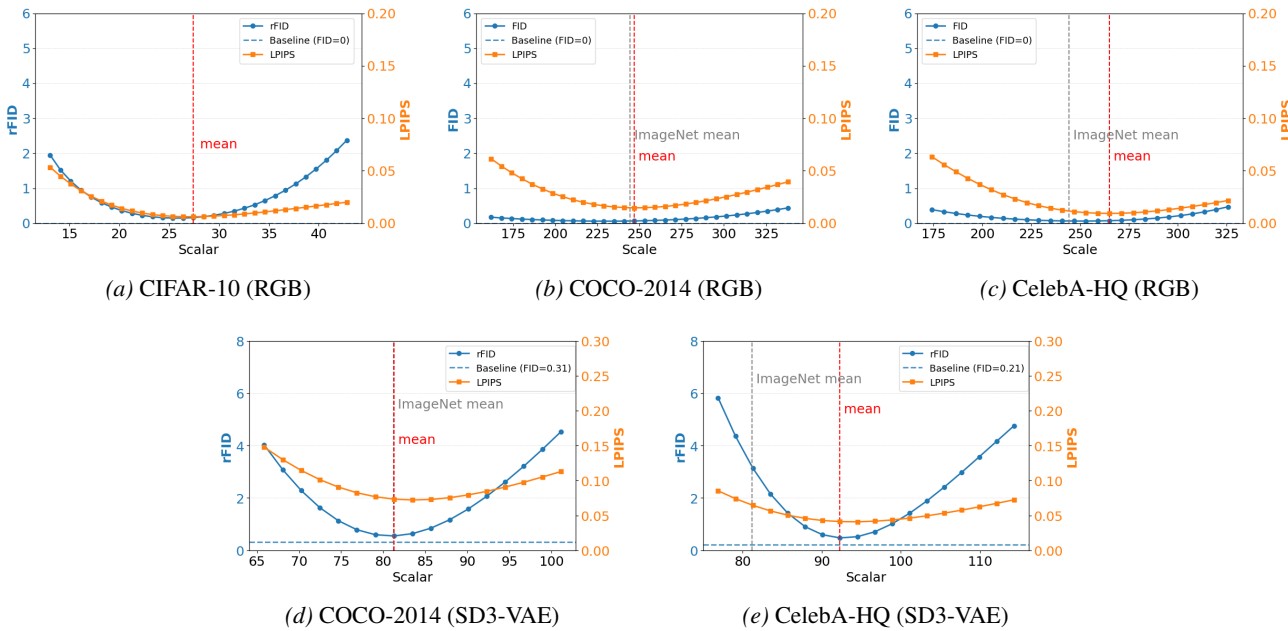

*(a)* CIFAR-10 (RGB)      *(b)* COCO-2014 (RGB)      *(c)* CelebA-HQ (RGB)

*(d)* COCO-2014 (SD3-VAE)      *(e)* CelebA-HQ (SD3-VAE)

*Figure II.* **Generalization of spherical projection across diverse datasets.** Reconstruction quality (rFID and LPIPS) with projection radius for various datasets. Top row: RGB space results for (a) CIFAR-10 (32×32), (b) COCO-2014, and (c) CelebA-HQ (both 256×256). Bottom row: SD3-VAE latent space results for (d) COCO-2014 and (e) CelebA-HQ. Vertical red dashed lines indicate the average norm of each dataset, and vertical gray dashed lines indicate the average norm of ImageNet.

for high-quality generation. The negligible improvement from explicit norm modeling suggests that directional components indeed dominate the semantic encoding, making our spherical approximation both theoretically sound and practically sufficient.

Figure I provides visual confirmation of the quantitative results. Given the minimal numerical improvements, it is unsurprising that generated samples appear almost identical across all three conditions (no refinement, ResNet50, MobileNetV2). We examined diverse ImageNet categories—from fine-grained textures (iguana scales, jewelry) to natural scene (volcano) and detailed subjects (hummingbird, pug)—yet found no visible differences.

| Model | Params (M) | GFLOPs | Inference (ms) |
|---|---|---|---|
| MobileNetV2 | 4.22 | 0.0043 | 6.70 |
| ResNet50 | 23.90 | 0.0251 | 5.26 |
| FM (per step) | 675.21 | 7.37 | 42.17 |
| FM (250 steps) | 675.21 | 1841.6 | 10540 |

*Table II.* **Computational cost comparison.** Computational cost comparison of the norm refinement networks and the main generation model. All measurements were obtained on an NVIDIA A100-40GB GPU.

## C.3. Computational Cost

We evaluated the computational requirements of norm refinement to ensure its overhead does not outweigh potential

benefits. All experiments were conducted on an NVIDIA A100-40GB GPU under the same benchmarking protocol used for the main generation model.

As summarized in Table II, both ResNet50 and MobileNetV2 show millisecond-level inference latency with very small computational footprints ($<$ 0.03 GFLOPs). These costs are effectively negligible relative to the diffusion model's per-step cost. For reference, training the MobileNetV2 regressor required 0.044 seconds per iteration, totaling roughly 3 hours for 10,000 iterations.

Taken together, these results show that while norm refinement is lightweight and inexpensive to deploy, its marginal benefits offer little justification for its addition. In practice, the hyperspherical projection using a single global average norm remains both a simpler and equally effective choice for high-quality generation. That said, if additional improvements are desired, a more sophisticated refinement module could be explored, as our current design uses only a minimal architecture and loss formulation, leaving room for potential enhancements.

## D. Additional Analysis

### D.1. Robustness of Projection Across Datasets

To verify the generality of our spherical projection approach, we evaluate reconstruction quality across varying projection radii on CIFAR-10 (Krizhevsky & Hinton, 2009), COCO-

| Dataset | RGB Space | | SD3-VAE Latent | |
|---|---|---|---|---|
| | Baseline | Mean Projected | Baseline | Mean Projected |
| CIFAR-10 | 0.00 | 0.47 (+0.47) | – | – |
| ImageNet-256 | 0.00 | 0.05 (+0.05) | 0.21 | 0.40 (+0.19) |
| COCO-2014 | 0.00 | 0.05 (+0.05) | 0.21 | 0.31 (+0.10) |
| CelebA-HQ | 0.00 | 0.01 (+0.01) | 0.21 | 0.21 (±0.00) |

*Table III*. **Mean-norm spherical projection across datasets.** Baseline shows unprojected rFID values; Mean Projected shows rFID after projecting each dataset to its mean norm. The minimal degradation confirms that average-norm projection preserves semantic information across diverse datasets. CIFAR-10 latent results are omitted as SD3-VAE is designed for higher-resolution inputs.

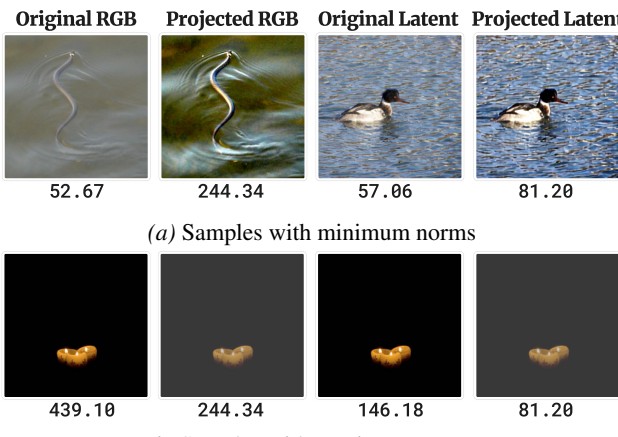

*(a)* Samples with minimum norms

*(b)* Samples with maximum norms

*Figure III*. **Visual impact of hyperspherical projection on extreme norm cases.** Samples with (a) minimum and (b) maximum L2 norms from ImageNet-256, selected independently per space. Note that maximum-norm samples coincide across RGB and SD3-VAE latent spaces, while minimum-norm samples differ, revealing distinct magnitude distributions. Unlike typical samples (Figure 1), extreme cases exhibit visible degradation after projection, yet semantic content remains preserved—confirming directional dominance in semantic encoding even at distribution boundaries.

2014 (Lin et al., 2014), and CelebA-HQ (Liu et al., 2015). Figure II shows that both RGB and latent spaces maintain good reconstruction quality near each dataset's mean norm (vertical red dased lines), confirming our approach works across diverse image domains.

Table III quantifies the reconstruction quality after mean-norm projection. The minimal rFID increases demonstrate that projecting to the dataset's own mean preserves semantic content effectively across all tested datasets.

COCO-2014, containing diverse object categories similar to ImageNet, has mean norms very close to ImageNet's in both RGB and latent space. Consequently, it maintains excellent reconstruction quality at ImageNet's mean radius in both representations. In contrast, CelebA-HQ—an extreme case where face images—exhibits distinctly different mean norms from ImageNet in both spaces.

Interestingly, despite this norm mismatch, CelebA-HQ still achieves good reconstruction at ImageNet's mean in RGB space, but suffers some degradation in latent space. This divergent behavior suggests that latent representations encode more semantic information through magnitude than RGB representations. However, since CelebA-HQ represents an extreme case—highly homogeneous images dominated by skin tones—such specialized datasets may require individual treatment. For the vast majority of natural image datasets with reasonable diversity, ImageNet's mean norm appears to serve as an effective universal radius.

### D.2. Extreme Norm Cases

We demonstrated in Section 3.1 that projection to average norm preserves visual quality in most cases. However, as shown in Table 1, information is not perfectly preserved, motivating us to investigate extreme norm cases. Figure III visualizes samples with the minimum and maximum norms from the ImageNet-256 validation set, projected onto the average norm. These extreme cases exhibit noticeable visual artifacts—particularly for minimum-norm samples, color casts in RGB space and brightness shifts in latent space

While directional components preserve most semantic information even at distribution boundaries, these extreme cases reveal that magnitude has a larger perceptual impact than it does for typical samples.

Importantly, Figure IV reveals that such extreme cases are rare outliers in the distribution. The RGB space shows a broad, symmetric distribution centered around 250, while the latent space exhibits a sharper, left-skewed distribution peaking near 80. The rarity of these boundary cases (less than 1% of data) confirms that hyperspherical projection remains viable for the vast majority of samples, with degradation limited to statistical outliers. While our approach remains robust, datasets with extreme luminosity shifts (e.g., day/night transitions) where magnitude is tied to semantics might exhibit minor limitations. Consequently, strictly fixing the norm could be suboptimal in these specific instances. Future research could address this challenge by further refining magnitude recovery strategies, where advancing our proposed norm prediction network offers a promising pathway.

### D.3. Class-Specific Norm Analysis.

While our spherical projection experiments demonstrate that a single global norm sufficiently preserves semantic content, we observe interesting class-specific patterns in the natural norm distributions. Figure V reveals that different ImageNet classes exhibit distinct norm characteristics, particularly in latent space. For instance, simple objects (e.g., "packet", "ring binder") tend to have lower latent norms compared

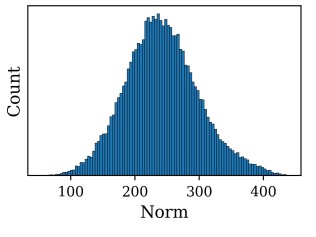

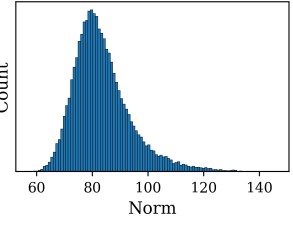

*(a)* RGB Space  *(b)* Latent Space

*Figure IV.* **Norm distribution of ImageNet-256 in RGB and latent spaces.** The distributions show distinct characteristics: RGB space exhibits a broad, symmetric distribution centered around 250, while latent space shows a sharp, left-skewed peak near 80 with a long tail. These different distributions explain why extreme cases behave differently across spaces.

| Method | Top-1 (%) | Top-5 (%) |
|---|---|---|
| *Reconstruction* | | |
| Original Images | 80.35 | 95.12 |
| DC-AE Recon | 77.30 | 93.67 |
| DC-AE Proj Recon | 76.29 | 92.97 |
| *Generation* | | |
| I-CFM | 61.83 | 80.71 |
| I-CFM Proj (Ours) | 79.53 | 93.51 |
| SFM (Ours) | **87.13** | **96.99** |

*Table IV.* **Downstream classification accuracy on ImageNet-256.** Top-1 and Top-5 accuracy of a pretrained ResNet-50 classifier evaluated on the original validation set (reconstruction rows) and on 50,000 generated samples (generation rows). DC-AE Proj Recon denotes images reconstructed from hyperspherically projected latents.

to complex natural scenes (*e.g.*, "accordion", "bib"). This suggests that while magnitude information is not critical for reconstruction, it may encode subtle semantic properties such as visual complexity or texture richness. However, the overlapping distributions and successful projection results confirm that these class-specific variations can be effectively approximated by a global average without significant perceptual loss.

### D.4. Downstream Classification of Reconstructed and Generated Images

We additionally evaluate semantic preservation and generation quality through the lens of a pretrained ResNet-50 classifier on ImageNet-256, reported in Table IV. This experiment serves two complementary purposes. First, the reconstruction rows assess whether hyperspherical projection preserves semantic content beyond what rFID and LPIPS capture, providing a class-discriminability perspective on our core geometric hypothesis. Second, the generation rows offer an alternative measure of generation quality that is independent of FID, further validating the superiority of our spherical methods.

**Reconstruction.** Comparing Original Images, DC-AE Recon, and DC-AE Proj Recon reveals that hyperspherical projection incurs only a marginal Top-1 accuracy drop of 1.01%p (77.30% → 76.29%) on top of the autoencoder's own reconstruction loss. This is consistent with and complementary to the rFID and LPIPS results in Tables 1 and III: class-discriminative features—which are arguably the most semantically meaningful—are robustly retained in the directional components even after norm normalization.

**Generation.** The generation rows tell a more striking story. Despite sharing the same architecture and training data, I-CFM achieves a Top-1 accuracy of only 61.83%, revealing that Euclidean flow matching produces samples with substantially degraded class discriminability. Incorporating hyperspherical projection alone (I-CFM + Proj) closes

much of this gap, recovering accuracy to 79.53%—nearly on par with the DC-AE autoencoder reconstruction itself—demonstrating that constraining the target to the hypersphere directly improves the semantic fidelity of generated samples. Most notably, SFM achieves a Top-1 accuracy of 87.13%, surpassing even the original validation set accuracy (80.35%). This result suggests that fully geometry-aware generative dynamics do not merely preserve class structure but actively enhance class discriminability, likely because geodesic flow paths on the hypersphere yield a more semantically consistent generative trajectory. Taken together, these results reinforce our central claim from two independent angles: hyperspherical projection faithfully preserves semantic content, and exploiting the intrinsic spherical geometry of natural images leads to meaningfully higher-quality generation beyond what FID alone can capture.

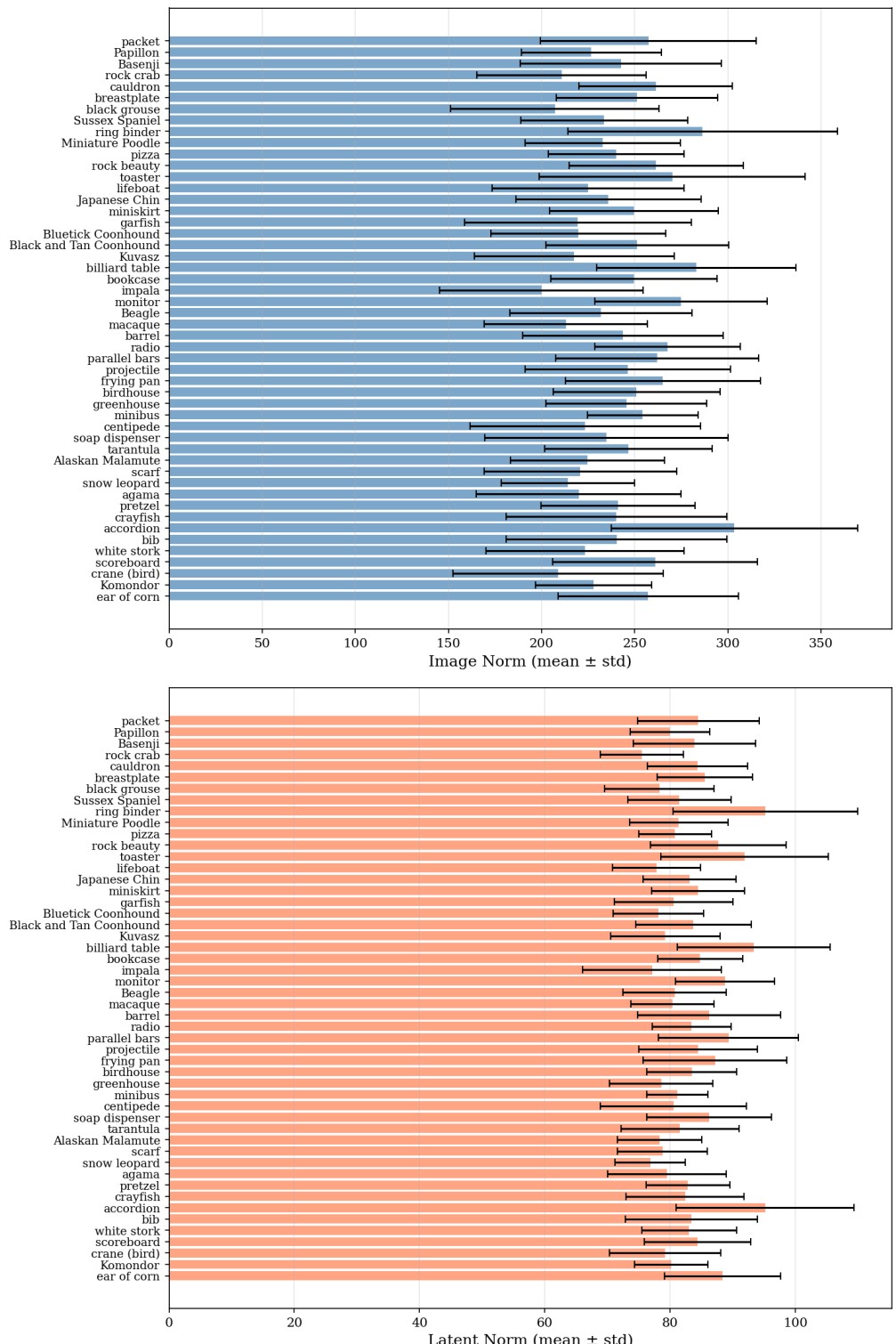

*Figure V.* **Class-specific norm distributions in RGB and latent spaces.** Mean L2 norms with standard deviations for 50 randomly selected ImageNet-256 classes. (Top) RGB space shows relatively uniform distributions across classes. (Bottom) Latent space (SD3-VAE) exhibits more diverse class-specific patterns, suggesting that different semantic categories can occupy distinct magnitude ranges in the latent representation.

