# OpenReview forum: "Geometry-Aware Image Flow Matching"
_ICML.cc/2026/Conference — ICML 2026 regular_

### Official Review · Reviewer_Bm7U · 2026-02-28

**Soundness:** 3
**Presentation:** 3
**Significance:** 4
**Originality:** 4
**Overall Recommendation:** 4
**Confidence:** 4

**Summary:**

This work investigates whether natural images exhibit intrinsic hyperspherical geometry, proposing that semantic content resides primarily in directional components while norms can be approximated by dataset averages. Based on this observation, the authors introduce two geometry-aware flow matching methods: SOT-CFM, which replaces Euclidean transport costs with angular metrics in optimal transport coupling, and SFM, which constrains both source/target distributions and flow dynamics directly on the hyperspherical manifold using geodesic paths

**Compliance With Llm Reviewing Policy:**

Affirmed.

**Final Justification:**

The rebuttal has substantially strengthened the manuscript by resolving empirical gaps, clarifying methodological boundaries, and providing deeper insights into optimization dynamics. The proposed geometric reformulation offers a principled, efficient, and empirically validated alternative to Euclidean flow matching. I therefore maintain my recommendation of Weak Accept, with the expectation that the final version will incorporate the promised revisions.

**Key Questions For Authors:**

see Weaknesses

**Limitations:**

yes

**Strengths And Weaknesses:**

# Strengths
• The core insight that directional components dominate semantic encoding is empirically validated across RGB and multiple latent spaces with rFID/LPIPS metrics.
• The mathematical formulation of spherical flow matching is rigorous, with closed-form geodesic interpolation and tangent vector derivation provided.
• Comprehensive ablation on norm refinement demonstrates that global average norm approximation suffices for high-quality generation, strengthening the practical viability of the approach.
• The work bridges Riemannian generative modeling and natural images, offering a novel perspective with potential impact on geometry-aware representation learning.

# Weaknesses
• The claim that "intrinsic manifold structure of natural images is largely unknown" is not directly measured; the paper demonstrates directional dominance but does not prove Euclidean methods fundamentally fail to capture this property.
• SOT-CFM's angular cost aims to prioritize semantic similarity, yet Table 2 shows minimal gains over OT-CFM with spherical projection (ImageNet-256 gFID: 5.18→5.15), suggesting the Euclidean cost may already suffice when norms are normalized.
• Results are reported on only two datasets (CIFAR-10, ImageNet-256) without cross-dataset generalization analysis beyond Appendix D.1; no ablation on hypersphere radius selection despite its critical role in projection.
• BClassifier-free guidance scales are "individually optimized for peak performance" per method, which may introduce tuning bias favoring proposed methods if search budgets differ.
• The paper acknowledges extreme norm cases show visible degradation after projection but does not quantify how frequently such cases occur in practice or propose mitigation strategies beyond optional norm refinement.

---

> ### Author Rebuttal · Authors · 2026-03-31
>
> Thank you for the feedback.
>
> __W1. Lack of evidence for Euclidean limitations__
>
> To clarify, we do not claim that Euclidean methods "fundamentally fail" to capture directional properties. As shown in Table 2, Euclidean-based methods already achieve reasonable performance. In Euclidean space, however, the model must optimize both direction and magnitude, even though semantic information is primarily encoded in direction, which can lead to suboptimal results. Our claim is that because the data manifold is highly concentrated along directional components, explicitly operating within a closed spherical space significantly eases the optimization difficulty. Consistent with this claim, Table 2 shows that applying projection consistently improves the performance across methods and datasets.
>
> __W2. Minimal gains over OT-CFM with spherical projection__
>
> We acknowledge that the performance gain of SOT-CFM over OT-CFM on ImageNet-256 appears modest (e.g., gFID 5.22 $\to$ 5.15). In high-dimensional latent spaces, Gaussian samples within a mini-batch are widely dispersed, making pairings induced by angular and Euclidean distances often similar. As a result, the choice of transport cost has a limited effect in this setting.
>
> Nevertheless, the angular metric shows more meaningful improvements on CIFAR-10 and consistently provides a slight edge across multiple runs on ImageNet. We therefore view SOT-CFM as a principled approach that aligns the optimal transport cost with the intrinsic semantic structure of the data.
>
> __W3. Ablation on hypersphere radius selection__
>
> We conducted additional ablation studies on the choice of hypersphere radius using both I-CFM with data projection and SFM:
>
> |Radius|I-CFM Proj|SFM|
> |:--:|:--:|:--:|
> |**10**|8.14|6.47|
> |**20**|6.18|5.38|
> |**30**|5.55|4.95|
> |**40**|5.17|4.87|
> |**45.25**|5.02|4.85|
> |**50**|5.11|4.78|
> |**60**|5.25|4.74|
> |**70**|5.56|4.70|
> |**80**|6.31|4.68|
>
> In I-CFM with data projection, we observe that performance improves as the projection radius approaches the average norm of Gaussian samples (45.25). This is because aligning the projection radius with average norm of gaussian samples minimizes unnecessary radial transport and allows the model to focus on learning semantically meaningful directional dynamics.
>
> In contrast, SFM projects both the source and target distributions onto a hypersphere with a fixed radius. While our default setting used the Gaussian average (45.25), these new results reveal that SFM's performance consistently improves as the radius increases. We hypothesize that a larger radius magnifies the tangent vectors for a given angular motion, thereby providing a stronger and clearer regression signal for the neural network in the tangent space.
> While we do not expect this improvement to scale indefinitely, we are currently running extended experiments to identify the optimal radius saturation point.
>
> We greatly thank the reviewer for the helpful suggestion, which led to an improved setting. We will incorporate these findings into the main manuscript.
>
> __W4. CFG scales are optimized individually__
>
> We assure the reviewer that the individual CFG optimization did not introduce any bias favoring our proposed methods. In fact, the base network and training hyperparameters were originally chosen following LightningDiT (I-CFM), instead of tuning specifically for our method.
>
> Regarding CFG, because different methods yield vector fields with distinct characteristics, applying a single, universal CFG scale would actually unfairly penalize one method by preventing the other from reaching their true maximum potential. Therefore, we optimized the CFG scale for each method within the exact same search budget specifically to ensure a fair, "peak-to-peak" performance comparison.
>
> __W5. Quantification and impact of extreme norm cases__
>
> We agree with the reviewer. To this end, we provide DINO similarity as a proxy for human perception under our setting. In particular, we observe that samples with DINO similarity below 0.7 are consistently perceived as extreme cases, but they are indeed rare (about 0.05%).
>
> If such cases had a meaningful impact on training, we would expect a degradation in generation quality (e.g., FID) or a mismatch between the norm distribution of generated samples and that of the data. However, neither of these is observed in practice. We consistently observe improved FID with projection across all settings, and the norm statistics of generated samples closely match those of the ImageNet validation set:
>
> |**Stats**|**Gen Img**|**Real Img**|
> |:--|:--:|:--:|
> |**mean**|58660.8|57449.4|
> |**std**|14164.9|13594.8|
> |**min**|5202.5|2960.4|
> |**25%**|49787.9|49283.9|
> |**50%**|57788.8|57119.7|
> |**75%**|66458.2|64956.3|
> |**max**|111528.4|111367.2|
>
> These results indicate that the model is not influenced by such rare extreme cases and instead captures the dominant data distribution. This may also explain why additional norm alignment techniques provide limited gains.

---

> > ### Author Rebuttal · Reviewer_Bm7U · 2026-04-03
> >
> > The authors have provided thoughtful responses and additional experiments that address several of my concerns. The clarification on Euclidean limitations (W1) helps reframe the contribution as optimization-focused rather than claiming fundamental failure of existing methods. The radius ablation study (W3) is valuable and reveals interesting behavior, particularly the finding that SFM benefits from larger radii. The quantification of extreme norm cases (W5) using DINO similarity and norm statistics is reassuring regarding practical impact.
> >
> > However, several core questions remain that are difficult to resolve within a rebuttal timeframe. First, while the spherical projection consistently improves results, the marginal gains from the angular metric in SOT-CFM (W2) remain modest in high-dimensional settings, raising questions about whether the added geometric complexity yields proportional benefits over simpler normalization strategies. Second, the radius sensitivity analysis (W3) uncovers a new consideration: SFM performance improves with larger radii, yet the theoretical justification for this trend and the identification of an optimal saturation point require further investigation beyond the current manuscript. Third, the generalization of the hyperspherical assumption to domains where magnitude may carry stronger semantic signal (e.g., medical imaging, remote sensing) remains unexplored, limiting confidence in the method's broader applicability.

---

> > > ### Author Response · Authors · 2026-04-04
> > >
> > > Thank you for the thoughtful and constructive feedback. We want to address each point below and clarify how our results support the main claims of the paper.
> > >
> > > __1. Marginal gains of SOT-CFM in high-dimensional settings__
> > >
> > > We agree that the incremental gain of SOT-CFM becomes modest in high-dimensional regimes, and we will revise the paper to better position this component of the contribution. Our results consistently show that the dominant improvement comes from hyperspherical projection itself, and more strongly from the fully geometry-aware SFM formulation. In particular, projection alone already provides stable gains across multiple settings, while SFM achieves the best overall performance.
> > >
> > > This behavior is also consistent with high-dimensional geometry. When samples are normalized or concentrated near a common norm, Euclidean and angular costs naturally induce increasingly similar transport pairings. From this perspective, the reduced gap between OT-CFM and SOT-CFM is expected rather than contradictory. At the same time, SOT-CFM remains meaningful because it aligns the transport objective with the same angular geometry imposed by the hyperspherical formulation.
> > >
> > > __2. Radius sensitivity and its interpretation__
> > >
> > > Our extended experiments reveal a clear and reproducible trend: performance improves with increasing radius and eventually saturates, with **the best performance observed a radius of 120 (FID: 4.62)** in our setting. Importantly, what matters is not a specific numeric value, but a regime in which tangent scaling improves optimization conditioning before reaching diminishing returns. Performance also remains stable across a broad range of nearby radii, indicating low sensitivity rather than reliance on a narrowly tuned value.
> > >
> > > This behavior can be interpreted through the geometry of the spherical formulation. Increasing the radius effectively scales the magnitude of tangent vectors along geodesic paths, which improves optimization conditioning and strengthens the training signal.
> > >
> > > Importantly, this is not merely a trivial scaling effect. In SFM, the vector field is defined intrinsically on the tangent space of the hypersphere, and thus the radius directly controls the magnitude of geodesic velocities while preserving the underlying angular structure. As a result, larger radii provide a better-conditioned learning problem specifically for directional dynamics, rather than simply rescaling the inputs.
> > >
> > > This explains the consistent improvement observed at larger radii, while the saturation reflects a regime where further scaling no longer provides additional optimization benefits.
> > >
> > > __3. Scope with respect to natural images__
> > >
> > > Our study focuses on natural-image distributions and their learned latent spaces. The consistency of our findings across multiple datasets and representation spaces indicates that the observed hyperspherical structure is not tied to a specific benchmark or encoder within this scope.
> > >
> > > For domains such as medical or scientific imaging, where magnitude may carry stronger domain-specific semantics, the underlying manifold structure may differ. We therefore view such domains as an important direction for future validation rather than a scope established in the current paper. We will clarify in the final version that our claims are specifically framed around natural-image distributions, rather than suggesting universality across all visual domains.
> > >
> > > Overall, we believe these clarifications directly address the core concerns. Our results consistently demonstrate that (i) hyperspherical projection is the primary source of improvement, (ii) the radius effect follows a clear and interpretable geometric trend rather than requiring delicate tuning, and (iii) the scope of our claims is well-defined around natural-image distributions. Taken together, these findings provide a coherent and well-supported explanation of the observed behavior, rather than isolated empirical observations.

---

### Official Review · Reviewer_qoW1 · 2026-03-08

**Soundness:** 2
**Presentation:** 3
**Significance:** 3
**Originality:** 3
**Overall Recommendation:** 4
**Confidence:** 3

**Summary:**

This paper decomposes natural images into directional components and norm components, and claims that most semantic information is encoded in the directional components, while the norm components can be approximated by the global average. Based on this finding, the paper proposes two methods: 1) SOT CFM,  which replaces Euclidean distances with angular distances in OT-CFM. 2) SFM, which projects the source and target samples to a hypersphere and trains the vector field along geodesic paths defined by spherical linear interpolation.

**Compliance With Llm Reviewing Policy:**

Affirmed.

**Final Justification:**

Thank you for the detailed response.

Compared to the improvement of 0.46 (5.25->4.79), the degradation caused by Gaussian Projection, 0.23 (5.00->5.23),  appears not slight. Since the method assumes that the RGB images, the Gaussian noise, and their interpolations all lie on a hypersphere, this raises my concerns about whether the assumption that the Gaussian noise and the interpolations lie on the hypersphere is reasonable.

However, the overall finding and the idea are interesting. Therefore, I tend to keep my score.

**Key Questions For Authors:**

See Weaknesses 1-4.
Are the real samples used in gFID the original unprojected images or their projected counterparts?

**Limitations:**

yes

**Strengths And Weaknesses:**

Strengths

1. The finding and the idea are interesting.
2. The experiments show improvements compared to OT-CFM and I-CFM.

Weaknesses
1. More evaluation metrics are needed in Table 1. The paper claims that most semantic information is encoded in the directional components. Therefore, it would be helpful to include semantic-level metrics, such as CLIP and DINO feature consistency and classification accuracy, to better support this claim. Moreover, standard reconstruction metrics like SSIM and PSNR would provide a more complete evaluation.
2. The impact of projecting the original Gaussian source distribution onto the hypersphere is not explicitly evaluated. Given that this is a central approximation in the method, its effect should be quantified.
3. The paper lacks qualitative comparisons with the baselines.
4. A comparison with other geometry-aware flow matching methods would strengthen the evaluation. For example, Metric Flow Matching (NeurIPS 2024) [1] is also a geometry-aware CFM method. The current paper approximates the image or latent manifold with a hypersphere, while Metric Flow Matching does not assume a known manifold structure. Comparing against it would help clarify the specific advantage of the proposed spherical formulation.

[1] Metric Flow Matching for Smooth Interpolations on the Data Manifold

---

> ### Author Rebuttal · Authors · 2026-03-31
>
> Thank you for your feedback.
>
> __W1. Include semantic-level and reconstruction metrics__
>
> We appreciate the reviewer’s constructive suggestion. Following the suggestion, we additionally provide CLIP and DINO feature consistency metrics, alongside traditional reconstruction metrics including PSNR and SSIM on the ImageNet-256 validation set below.
>
>
> |Space|PSNR↑(Base/Proj)|SSIM↑(Base/Proj)|DINO↑(Base/Proj)|CLIP↑(Base/Proj)|
> |:--|:--:|:--:|:--:|:--:|
> |SD2-VAE|26.91/25.36|0.737/0.717|0.941/0.926|0.970/0.957|
> |SD3-VAE|29.59/27.39|0.877/0.859|0.981/0.971|0.988/0.981|
> |VMAE|31.52/30.31|0.890/0.877|0.972/0.968|0.980/0.979|
> |DC-AE|23.27/22.54|0.654/0.640|0.915/0.897|0.963/0.954|
>
> "Base" denotes the standard autoencoder reconstruction, whereas "Proj" indicates our hyperspherical projection to the encoded latent representation before decoding. Both reconstructed outputs are compared directly against the original, unprojected images. As shown in the table above, our spherical models achieve highly competitive and robust similarity across all metrics.
>
> Notably, while spherical projection causes minor, expected drops in pixel-level metrics (PSNR and SSIM), semantic-level metrics (DINO and CLIP) remain exceptionally high. This aligns with our core hypothesis: essential semantic information is robustly preserved in the directional components. We will include these results in the final manuscript.
>
> __W2. Impact of projecting original Gaussian source onto the hypersphere__
>
> To clarify, projecting the source distribution onto the hypersphere is not a central contribution of our work. Its role in SFM is simply to ensure that both the source and target lie on the same manifold, enabling flow matching to be performed consistently within the hyperspherical geometry.
>
> In fact, for I-CFM, projecting both source and target onto the same hypersphere results in comparable performance. We hypothesize that this is because high-dimensional Gaussian distributions are already tightly concentrated on a thin spherical shell (Sec. 3.3), making explicit projection of the source largely redundant.
>
> Our main contributions are twofold: (i) the observation that the target data manifold can be effectively approximated as a hypersphere, and (ii) a geometry-aware flow matching formulation defined on this hyperspherical manifold.
>
> As shown in Table 2, projecting the target distribution consistently improves performance across multiple flow matching methods, demonstrating the benefit of modeling data on the hypersphere. Furthermore, our geometry-aware formulation (SFM) builds on this and yields additional improvements.
>
> __W3. Missing qualitative comparisons__
>
> We appreciate the feedback. We will include qualitative comparisons and visual examples in the final manuscript.
>
> __W4. Comparison with Metric Flow Matching (MFM)__
>
> We thank the reviewer for pointing out MFM. We will add a detailed discussion of MFM to our related work section.
>
> While MFM provides a powerful, generalized framework for learning flows by implicitly learning the underlying Riemannian metric without assuming a known manifold, this generality naturally comes at the cost of computational overhead and reliance on metric approximations during training. Our approach, however, stems from a fundamentally different starting point and offers distinct structural advantages.
>
> By empirically discovering and demonstrating that the manifold of natural image data can be effectively approximated as a hypersphere, we completely bypass the need to implicitly learn the data geometry. The specific advantage of our spherical formulation lies in its mathematical exactness and computational efficiency. Because the hyperspherical manifold is explicitly known, our method grants direct access to strictly defined, closed-form geodesics and exact tangent spaces.
>
> Therefore, rather than requiring the model to allocate representational capacity and compute to approximate the metric constraints, our method explicitly leverages a fixed, perfectly defined geometry. Thus, rather than acting as a direct competitor to the generalized MFM framework, our approach establishes an empirically validated geometric prior that provides an efficient flow matching formulation tailored specifically for natural images.
>
> __Q1. Are the real samples used in gFID the original unprojected images or their projected counterparts?__
>
> To ensure a completely fair and rigorous evaluation, all gFID metrics reported for our proposed methods and baselines were calculated against the original, unprojected real images.

---

> > ### Author Rebuttal · Reviewer_qoW1 · 2026-04-01
> >
> > Thanks for the reply. Some of my concerns have not been addressed.
> > - W1: Would the drop in the pixel-level metrics influence the application of the downstream task?  Could you provide the classification accuracy between the generated images of your method and baselines without the  projection? Could you also provide the classification accuracy between the original images and the projected images?
> > - W2: The work assumes the RGB images and the Gaussian noise lying on the hypersphere. However,  it only evaluates the impact of projecting RGB images onto the hypersphere, but does not evaluate the impact of projecting Gaussian noise onto the hypersphere. Could you quantify the impact of this approximation?
> > - W3: Can you provide the qualitative comparisons?
> > - W4: Can you provide the qualitative and quantitative comparison to show the statement 'The specific advantage of our spherical formulation lies in its mathematical exactness and computational efficiency'? Why does the spherical formulation provide better mathematical exactness?

---

> > > ### Author Response · Authors · 2026-04-07
> > >
> > > Thank you for the thoughtful feedback, and we apologize for the delayed response. Ensuring an accurate implementation of MFM required additional time.
> > >
> > > __[W1] Impact of Projection on Downstream Tasks__
> > >
> > > We evaluated downstream task performance using ImageNet-256 classification with a pretrained ResNet-50.
> > >
> > > **Reconstruction Classification (Val Set)**
> > > | Method | Top-1 Acc (%) | Top-5 Acc (%) |
> > > |--|--|--|
> > > | Original Images | 80.35 | 95.12 |
> > > | DC-AE Recon | 77.30 | 93.67 |
> > > | DC-AE Proj Recon | 76.29 | 92.97 |
> > >
> > > The table shows that hyperspherical projection causes only a **marginal Top-1 accuracy drop (1.01%p)**.
> > >
> > > **Generation Classification**
> > >
> > > | Method | Top-1 Acc (%) | Top-5 Acc (%) |
> > > |--|--|--|
> > > | I-CFM | 61.83 | 80.71 |
> > > | **I-CFM Proj (Ours)** | **79.53** | **93.51** |
> > > | **SFM (Ours)** | **87.13** | **96.99** |
> > >
> > > Furthermore, evaluating the generated samples reveals that our SFM and I-CFM + Projection achieve the highest and second-highest accuracy, respectively. These results are significantly better than those of the baseline I-CFM and even surpass the accuracy of the original validation set. This demonstrates that our geometry-aware formulation not only preserves but also enhances the semantic fidelity and class discriminability of the generated samples.
> > >
> > > __[W2] Impact of Gaussian Projection__
> > >
> > > As suggested, we evaluated the projection of the Gaussian source ($\tilde{\mathcal{N}}$).
> > >
> > > | Method | Source | Target | FID |
> > > |--|--|--|--|
> > > | I-CFM | $\\mathcal{N}$ | $\\mathcal{D}$ | 5.25±0.035 |
> > > | I-CFM | $\\mathcal{N}$ | $\tilde{\\mathcal{D}}$ | 5.00±0.020 |
> > > | **I-CFM (new)** | **$\tilde{\\mathcal{N}}$** | **$\tilde{\\mathcal{D}}$** | **5.23±0.020** |
> > > | SFM | $\tilde{\\mathcal{N}}$ | $\tilde{\\mathcal{D}}$ | 4.79±0.021 |
> > >
> > > Results show that projecting both source and target yields slightly worse FID than projecting only the data. This confirms that the performance gains of SFM do not stem from Gaussian projection.
> > >
> > > __[W3] Qualitative Comparison__
> > >
> > > The requested visual comparisons can be found at: https://anonymous.4open.science/r/Geometry-Aware-Image-Flow-Matching-9F10/README.md.
> > >
> > > We provide qualitative comparisons across three methods: I-CFM, I-CFM + Proj (Ours), and SFM (Ours). All methods share the same random seed and class labels.
> > >
> > > As shown in the figure, I-CFM frequently produces images with noticeable artifacts, incoherent structures, and blurred details. Adding hyperspherical projection to the target (I-CFM + Projection) yields moderate improvements in overall coherence. However, our geometry-aware formulation (SFM) consistently generates images with significantly sharper fine-grained details, more coherent global structures, and richer textures.
> > >
> > > __[W4] Comparison with MFM__
> > >
> > > First, we would like to clarify that our previous explanation may have been ambiguous. What we intended to convey is that, under our formulation where image data is approximated on a hypersphere, the definition of transport paths admits **mathematically exact (closed-form) geodesics under the assumed hyperspherical geometry**. In contrast, MFM relies on a learned geopath network to approximate the underlying geometry, which introduces additional approximation error and training complexity.
> > >
> > > That said, as the reviewer correctly pointed out, MFM can indeed be considered a relevant comparison to our approach. So, we conducted additional experiments.
> > >
> > > | | Method | FID ($\\downarrow$) |
> > > |---|---|---:|
> > > | Euclidean | I-CFM | 5.29 |
> > > |  | I-CFM Proj | 5.02 |
> > > | Geometry-Aware | **MFM** | 14.12 |
> > > |  | SFM | 4.81 |
> > >
> > > MFM shows inferior performance not only compared to our method, but also relative to the standard I-CFM baseline. This is because it is designed to **interpolate unknown distributions on learned manifolds** rather than Gaussian-to-data transport. Consequently, its dynamics are not optimized for noise-to-data generation. This limitation is evident in the MFM paper, which focuses on low-dimensional synthetic data rather than high-dimensional image benchmarks like ImageNet or CIFAR-10.
> > >
> > > In addition, MFM introduces significant overhead through a two-stage process: an **additional geodesic path learning stage** and **multiple time-derivative computations** during training. The computational comparison results are shown below.
> > >
> > > | | |Stage 1 |Stage 2 |Total|
> > > |--|--|--|--|--|
> > > |Memory/GPU|MFM|25.21 GiB|22.61 GiB|peak 25.21 GiB|
> > > ||SFM|0|20.93 GiB|peak 20.93 GiB|
> > > |Time/step|MFM|2.605 s|1.320 s|--|
> > > ||SFM|0|0.608 s|--|
> > > |Training Time|MFM|36.18 h|73.33 h|109.51 h|
> > > ||SFM|0|33.77 h|33.77 h|
> > >
> > > We use LightningDiT-L/2 for the geopath network of MFM and LightningDiT-XL/1 for both flow models. These results show that MFM incurs **much higher training cost**, due to the additional geopath learning stage and repeated derivative computations, while our method avoids these overheads by operating on a fixed geometry.
> > >
> > > We will include a detailed discussion of MFM and its differences from our approach in the final version of the paper.

---

### Official Review · Reviewer_aZmW · 2026-03-09

**Soundness:** 3
**Presentation:** 4
**Significance:** 3
**Originality:** 3
**Overall Recommendation:** 4
**Confidence:** 5

**Summary:**

This paper argues that natural images can be effectively modeled on a hypersphere, and proposes two approaches to demonstrate the same.

**Compliance With Llm Reviewing Policy:**

Affirmed.

**Final Justification:**

The authors did a good rebuttal. Most of my concerns have been resolved.

**Key Questions For Authors:**

Here are a few suggestions that I think would strengthen the paper:


1. It would also be helpful to show training curves, for example FID versus training time or epochs for a baseline and the proposed method. If the authors’ intuition is that the spherical projection simplifies learning, then faster or more stable convergence would be a convincing piece of evidence.

2. I would like to see stronger empirical support for the claim that the proposed approach is genuinely better. Reporting results across multiple seeds with error bars would help determine whether the gains in Table 2 are statistically meaningful.

3. Another useful support would be to prove whether the proposed method improves optimization properties, such as reducing gradient variance or stabilizing transport pairings, compared to the Euclidean baseline. At present, the paper argues for a simpler learning problem, but this is not directly demonstrated.

**Limitations:**

A key limitation is that the paper’s evidence is restricted to standard natural-image benchmarks and latent spaces derived from them, so it is unclear whether the same geometric hypothesis would hold for other visual domains, such as medical imaging, scientific imaging, or other image distributions.

**Strengths And Weaknesses:**

## Strengths

1. The paper presents an interesting and intuitive idea. I found the observation in Figure 1 particularly surprising: projecting images or latents to a hypersphere appears to preserve visual semantics quite well despite noticeable changes in norm.
2. The paper is clearly written and easy to follow. I appreciated the overall organization and presentation. Many of the natural questions about the method are answered in the text.

## Weakness

1. I am not yet convinced that the paper provides a strong explanation for why the proposed approach should help as much as claimed. At a high level, the method seems to reduce the problem from modeling data in $\mathbb{R}^{d}$ space to modeling data on $d-1$ dimensional subspace by discarding or fixing the norm component. While this difference would be substantial when $d$ is small, in very high dimensions, this change may not be substantial enough on its own to explain a meaningful gain in generation quality. The paper’s main support for this claim comes from sample-based metrics. However, the reported improvements over Euclidean baselines in Table 2 are fairly modest, so it is difficult to rule out the possibility that some of the gains are within the variance of training and sampling.
2. Some recent baselines are missing, especially stronger one-step generative methods. These are relevant comparisons because they target strong generation quality with much faster sampling. In contrast, the proposed method uses Euler sampling with 100 function evaluations on CIFAR-10 and 250 on ImageNet-256. I would like the authors to clarify why such baselines were not included.

[a] Inductive Moment Matching by Linqi Zhou, Stefano Ermon, Jiaming Song

[b] Mean Flows for One-step Generative Modeling by Zhengyang Geng, Mingyang Deng, Xingjian Bai, J. Zico Kolter, Kaiming He

---

> ### Author Rebuttal · Authors · 2026-03-31
>
> Thank you for your feedback.
>
> __W1. Clarification and additional evidence (Q1-3)__
>
> We thank the reviewer for the insightful comments and constructive suggestions.
>
> We emphasize that the benefit of our method does not arise from a simple dimensionality reduction from $\mathbb{R}^d$ to a $(d-1)$-dimensional subspace. Instead, the key change is geometric: by projecting onto a hypersphere, we move from an unbounded Euclidean space to a compact manifold with bounded support, which simplifies the learning problem.
>
> To provide deeper insights beyond final performance and directly address the reviewer’s suggestions, we analyzed our method across three key dimensions: training dynamics, statistical reliability, and gradient analysis.
>
> - **Training dynamics:** We compare FID of I-CFM, I-CFM with projection, and SFM under the same pairing setting. Projection and SFM consistently show faster and more stable convergence.
> |Step|I-CFM|I-CFM Proj (Ours)|SFM (Ours)|
> |:--:|:--:|:--:|:--:|
> |**20k**|79.94|*73.76*|**73.29**|
> |**40k**|*17.18*|17.69|**15.96**|
> |**60k**|9.03|*8.76*|**8.16**|
> |**80k**|6.93|*6.63*|**6.30**|
> |**100k**|6.03|*5.82*|**5.55**|
> |**120k**|5.59|*5.38*|**5.16**|
> |**140k**|5.29|*5.02*|**4.81**|
>
> - **Statistical reliability:** We report mean and standard deviation over three runs, confirming that our SFM outperforms all other approaches statistically significantly.
> |**Method**|**FID**|
> |:--|:--:|
> |**I-CFM**|5.25±0.04|
> |**I-CFM Proj (Ours)**|5.00±0.02|
> |||
> |**OT-CFM**|5.14±0.06|
> |**SOT-CFM (Ours)**|5.08±0.06|
> |**SOT-CFM Proj (Ours)**|5.04±0.05|
> |||
> |**SFM (Ours)**|**4.79±0.02**|
>
> - **Gradient analysis:** We evaluate optimization stability using scale-normalized gradient variance ($CV^2 = \text{Var}(\|g\|)/\mathbb{E}[\|g\|]^2$) and direction consistency (cosine similarity). Our methods improve both in early training:
>
> **1) Gradient variance ($CV^2$)** $\downarrow$
> > * **I-CFM:** 0.3549
> > * **I-CFM Proj (Ours):** 0.2941
> > * **SFM (Ours):** **0.2095**
>
> **2) Gradient direction consistency** $\uparrow$
> > * **I-CFM:** -0.0166
> > * **I-CFM Proj (Ours):** **0.1862**
> > * **SFM (Ours):** 0.0302
>
> Taken together, these results provide consistent evidence that our method improves convergence speed, stability, and final performance. We will include these analyses in the revised manuscript.
>
> __W2. Missing one-step baselines and sampling efficiency__
>
> We thank the reviewer for highlighting the recent 1-step generative models (IMM [a], Mean Flows [b]).
>
> Fundamentally, our objective is to validate the benefits of spherical geometry, not to compete directly with specialized 1-step frameworks. Methods like [a] and [b] design novel training objectives specifically to achieve 1-step generation. In contrast, our work geometrically redefines the underlying data manifold and flow trajectories. Because changing the geometric space of flow trajectory is orthogonal to modifying the training objective, these techniques can be seamlessly integrated.
>
> While fully training these models to convergence requires approximately 12 days on our setting, we are actively exploring their integration. Our preliminary integration with Mean Flows [b] at just 30K iterations clearly confirms this synergy: SFM + Mean Flow achieves a 1-step FID of 72.02, outperforming the Euclidean baseline (I-CFM + Mean Flow, FID 79.88).
>
> Furthermore, we agree that sampling efficiency is a crucial metric. Since Euclidean approaches like OT-CFM aim to construct straighter transport paths to reduce the Number of Function Evaluations (NFE), it is vital to verify if our geometry inherently enhances this property. To evaluate this without relying on specialized 1-step training, we conducted additional experiments at low NFEs and introduced OT-SFM, a strict spherical counterpart to Euclidean OT-CFM.
>
> | NFE | OT-CFM | OT-SFM (Ours) | I-CFM | SFM (Ours) |
> | :---: | :---: | :---: | :---: | :---: |
> | **1** | 256.44 | **166.54** | 256.44 | **190.24** |
> | **3** | 135.07 | **29.30** | 174.95 | **35.20** |
> | **5** | 37.80 | **11.94** | 54.14 | **14.17** |
> | **10** | 7.70 | **5.88** | 9.03 | **6.42** |
> | **20** | 5.67 | **5.12** | 6.05 | **5.36** |
> | **40** | 5.31 | **4.84** | 5.50 | **5.04** |
> | **60** | 5.27 | **4.76** | 5.41 | **4.95** |
> | **80** | 5.20 | **4.73** | 5.34 | **4.91** |
> | **100** | 5.23 | **4.69** | 5.34 | **4.87** |
>
>
> At low NFEs (1–10), our spherical methods reduce FID by up to ~70–80% compared to Euclidean baselines, while maintaining consistent improvements across all NFEs.
>
> These results indicate that hyperspherical geometry natively leads to straighter and more stable transport, enabling efficient sampling even without specialized one-step objectives. We will include the preliminary 1-step integration results, the new OT-SFM formulation, and the comprehensive low-NFE analysis in the revised manuscript.

---

> > ### Author Rebuttal · Reviewer_aZmW · 2026-04-03
> >
> > My issues have been resolved. Thank you and best of luck.

---

> > > ### Author Response · Authors · 2026-04-04
> > >
> > > We sincerely thank Reviewer aZmW for the positive feedback. Your constructive suggestions on optimization stability and low-NFE efficiency have significantly helped strengthen our paper.

---

### Official Review · Reviewer_ZmCo · 2026-03-13

**Soundness:** 2
**Presentation:** 3
**Significance:** 2
**Originality:** 2
**Overall Recommendation:** 3
**Confidence:** 4

**Summary:**

This paper first performs some systematic analysis, demonstrating that the directional components rather than the norms encode most significant semantic information. Based on this observation, the paper extends the existing Flow Matching based method and proposes a new SOT-CFM (Spherical Optimal Transport Flow Matching) method. The paper shows improved performance, compared with existing flow matching based methods.

**Compliance With Llm Reviewing Policy:**

Affirmed.

**Final Justification:**

I appreciate authors' effort in trying to clarify the position of the paper. However, compared with significant (similar) findings in previous work (with some important paper uncited), I think this greatly affects the novelty of the proposed methodology. Without this, the proposed method also does not demonstrate state-of-the-art performance. The method is built on a (not really new) observation, with limited improvements compared with previous angle-based methods, and without demonstration of state-of-the-art performance, or sufficient visual examples. I don't think the paper is sufficient for acceptance.

**Key Questions For Authors:**

How is this idea significantly different from previous work?

**Limitations:**

Limitations of the paper should be clearly stated.

**Strengths And Weaknesses:**

Strengths:
- The paper shows decent experimental results to support the observation. However, the improvements compared with angular-based methods are quite limited.
 - The proposed algorithm design is well motivated with an observation. However, the observation is only based on statistical analysis, rather than having a strong theoretical basis.

Weaknesses:
- The idea that semantics is largely encoded in the angular direction is not new. In fact, the widely used cosine similarity is an indication of the direction importance. There is a recent paper that tries to use direction alignment in diffusion based generation, which is relevant (but not cited in the current submission):

Angle Domain Guidance: Latent Diffusion Requires Rotation Rather Than Extrapolation, ICML 2025.

Please note that the hyperspherical domain is a natural derivative from the angle domain guidance (and other works that use cosine  similarity of angles). If the norm is not important for semantics, setting the norm to 1 should not lose semantics. Compared with angle-based methods, the improvements of the proposed method are limited.

- The paper only compares with Flow Matching methods, and does not demonstrate if the results are state-of-the-art.

- The testing datasets are quite small, and the paper does not show sufficient visual examples that results are clearly better.

---

> ### Author Rebuttal · Authors · 2026-03-31
>
> Thank you for your feedback.
>
> __W1. Limited novelty regarding angular direction & a missing reference (ADG)__
>
> We thank the reviewer for pointing out a relevant work, ADG. We agree that prior studies have shown that semantic information is largely encoded in angular directions. However, our contribution goes beyond simply leveraging angular similarity. To the best of our knowledge, this is the first work to show that the image latent space can be effectively approximated by a hyperspherical manifold even when the norms of all samples are approximated by a single global value, leaving the semantic and perceptual content almost intact. Based on this observation, we further develop a geometric formulation that explicitly operates on this hypersphere.
>
> In this regard, our approach differs from ADG in both scope and formulation. ADG uses angular alignment as a guidance mechanism during diffusion sampling, whereas we model the underlying data geometry itself. Specifically, we reformulate Flow Matching in a Riemannian setting (SFM), where trajectories and vector fields are defined directly on the hypersphere. We will include this comparison and discussion about ADG in the final version.
>
> __W2. Compared only with flow matching__
>
> We clarify that the primary objective of this work is _not_ to beat all existing generative modeling paradigms, but rather to identify an effective geometric approximation of the underlying manifold of natural images and study its impact on generative modeling. We focus on flow matching mainly because it allows flexible choices of the source distribution, enabling us to project both source and target onto the same hyperspherical manifold and study fully geometry-aware dynamics (SFM), which is difficult to achieve in standard diffusion frameworks.
>
> Importantly, our findings are not limited to this specific setting. As shown in Table 2, projecting only the target onto the hypersphere consistently improves performance, even with a standard Gaussian source. This suggests that our method is complementary to existing generative paradigms (e.g., diffusion), and can be naturally extended or combined with them in future work.
>
> __W3. Limited qualitative samples__
>
> We appreciate the feedback. We will include qualitative comparisons and visual examples in the final manuscript to clearly demonstrate the visual improvements.
>
> __Q. Key difference from previous work__
>
> _1. Natural Image Data Geometry (Sec. 3.1)_: To the best of our knowledge, this is the first work to empirically demonstrate that the manifold of natural image data can be effectively approximated by a hyperspherical geometry.
> We further show that this hyperspherical approximation simplifies the learning process and leads to consistent improvements in generative performance.
>
> _2. Training Objective (Sec. 3.2-3.3)_: Based on this geometric insight, we modify the flow matching formulation itself, beyond prior work that only uses angular information as a similarity measure:
>
> - SOT-CFM replaces the Euclidean optimal transport cost with an angular distance based cost for pairing.
> - SFM goes a step further by formulating both the trajectory (geodesics) and the objective function (Riemannian flow matching) directly on the sphere. To the best of our knowledge, SFM is the first geometry-aware flow matching method for image generation that learns directly on a hyperspherical manifold.
>
> _3. Scope of modification_: Unlike prior approaches that merely incorporate directional information at the similarity or sampling level, our method introduces a geometric formulation at the training level, effectively changing the space in which learning occurs.
>
> We hope this clarifies the distinctions of our work. If any part remains unclear, we would be happy to further discuss during the discussion period.

---

> > ### Author Rebuttal · Reviewer_ZmCo · 2026-04-03
> >
> > I appreciate authors' effort in trying to clarify the position of the paper. However, compared with significant (similar) findings in previous work (with some important paper uncited), I think this greatly affects the novelty of the proposed methodology. Without this, the proposed method also does not demonstrate state-of-the-art performance. The method is built on a (not really new) observation, with limited improvements compared with previous angle-based methods, and without demonstration of state-of-the-art performance, or sufficient visual examples. I don't think the rebuttal reasonably addressed my concerns, so I will keep my Weak Reject rating.

---

> > > ### Author Response · Authors · 2026-04-05
> > >
> > > We thank the reviewer for the feedback.
> > >
> > > ---
> > >
> > > The requested visual comparisons can be found at: https://anonymous.4open.science/r/Geometry-Aware-Image-Flow-Matching-9F10/README.md
> > >
> > > We provide side-by-side qualitative comparisons across three methods: I-CFM, I-CFM + Projection, and SFM (Ours). All methods share the same random seed and class labels, so each row shows how the same noise input is transformed into an image under each formulation.
> > >
> > >   As shown in the figure, I-CFM frequently produces images with noticeable artifacts, incoherent structures, and blurred details. Adding hyperspherical projection to the target (I-CFM + Projection) yields moderate improvements in overall coherence. However, our geometry-aware formulation (SFM) consistently generates images with significantly sharper fine-grained details, more coherent global structures, and richer textures. These qualitative improvements are consistent with the quantitative gains reported in Table 2, confirming that operating directly on the hyperspherical manifold leads to higher-fidelity generation.
> > >
> > > ---
> > >
> > > We respectfully disagree with the reviewer’s opinion about the novelty and scope of our contribution. Our reasoning is as follows.
> > >
> > > **1. Fundamentally new observation and method**
> > >
> > > First of all, our core observation is not simply that "direction matters." Rather, we show that natural image data—including high-dimensional latent representations—can be effectively approximated by a **hyperspherical manifold with a single global norm**, while preserving semantic and perceptual content. This is not merely a heuristic based on directional similarity, but a structural property that directly enables a **geometry-aware training formulation** for generative modeling.
> > >
> > > **We are open for further discussion if the reviewer provides any existing prior work that demonstrates both:**
> > > * **(i)** hyperspherical approximation of natural image or latent distributions using a global norm, and
> > > * **(ii)** a corresponding training formulation built upon this property in natural image.
> > > To the best of our knowledge, our work is the first one proposing both of these.
> > >
> > > ---
> > >
> > > **2. Mischaracterization of our contribution**
> > >
> > > We believe that characterizing our method as an “angle-based” approach substantially underestimates the scope of our work. While SOT-CFM incorporates an angular cost, it is only one component of our framework and is not the primary contribution.
> > >
> > > Our primary contribution is the **hyperspherical projection**, which removes unessential norm modeling and stabilizes training by simplifying the learning problem (see aZmW W1 for detailed evidence). In addition, we propose **SFM**, which performs flow matching directly on the hyperspherical manifold using geometry-aware dynamics.
> > >
> > > In short, the novelty of our work lies in **changing the training space and learning dynamics themselves**, rather than simply reusing angular similarity.
> > >
> > > ---
> > >
> > > **3. Incorrect comparison to prior work**
> > >
> > > To the best of our knowledge, even among existing angle-based methods, none modifies the **training formulation itself** in the way we do.
> > >
> > > The work cited by the reviewer (ADG) applies angular alignment as a **guidance mechanism during diffusion sampling** of an already trained model. In contrast, our method introduces a **hyperspherical geometric prior** and reformulates the training objective and dynamics accordingly. In other words, ADG is an **inference-time** sampling strategy, whereas our method directly modifies the **training mechanism** of the generative model.
> > >
> > > ---
> > >
> > > **4. Underestimation of empirical improvements**
> > >
> > > We have provided systematic and consistent empirical evidence that supports our claim. As shown in Table 2 of the main paper, simply projecting the target data onto the hypersphere (hyperspherical projection) already leads to consistent performance improvements across flow matching variants. Furthermore, our SFM achieves additional gains by learning geometry-aware dynamics on the sphere (e.g., FID $5.29 \rightarrow 4.81$). These improvements are statistically significant and consistent, directly supporting our main claim that hyperspherical geometry simplifies and improves generative learning.
> > >
> > > ---
> > >
> > > **Summary**
> > >
> > > In summary, we show that image data manifolds can be effectively approximated by a hypersphere, and we introduce a **model-agnostic framework** that improves generative learning by changing the training space and learning dynamics. We believe this constitutes a clearly novel contribution, both conceptually and empirically.

---

### Decision · Program_Chairs · 2026-04-30

**Decision:**

Accept (regular)

**Comment:**

The submission received 3 WA's and 1 WR. Reviewers aZmW and qoW1 found the authors' observations, findings and ideas to be interesting, with reviewer Bm7U also considering that the mathematical formulation is rigorous. There were some concerns that claims in the paper are not that well established theoretically or experimentally, and pre-rebuttal there were recommendations to undertake further experiments in terms of comparisons to additional baselines and use of other metrics. The responses were rigorous, with reviewers accepting many concerns they had were resolved, but not all. Nonetheless, the three more active reviewers were all leaning towards recommending accept, and the AC concurs with them.